# Rethinking MUSHRA: Addressing Modern Challenges in Text-to-Speech Evaluation

**Praveen Srinivasa Varadhan**[1]**, Amogh Gulati**[2]**, Ashwin Sankar**[1]**, Srija Anand**[1]**, Anirudh Gupta**[2]**, Anirudh Mukherjee**[2]**, Shiva Kumar Marepally**[1]**, Ankur Bhatia**[2]**, Saloni Jaju**[2]**, Suvrat Bhooshan**[2]**, Mitesh M. Khapra**[1]                 *{cs21d201, miteshk}@cse.iitm.ac.in*
[1]*AI4Bharat, Indian Institute of Technology Madras,* [2]*Gan.AI .*

**Reviewed on OpenReview:** *https://openreview.net/forum?id=oYmRiWCQ1W*

## Abstract

Despite rapid advancements in TTS models, a consistent and robust human evaluation framework is still lacking. For example, MOS tests fail to differentiate between similar models, and CMOS's pairwise comparisons are time-intensive. The MUSHRA test is a promising alternative for evaluating multiple TTS systems simultaneously, but in this work we show that its reliance on matching human reference speech unduly penalises the scores of modern TTS systems that can exceed human speech quality. More specifically, we conduct a comprehensive assessment of the MUSHRA test, focusing on its sensitivity to factors such as rater variability, listener fatigue, and reference bias. Based on our extensive evaluation involving 492 human listeners across Hindi and Tamil we identify two primary shortcomings: (i) *reference-matching bias*, where raters are unduly influenced by the human reference, and (ii) *judgement ambiguity*, arising from a lack of clear fine-grained guidelines. To address these issues, we propose two refined variants of the MUSHRA test. The first variant enables fairer ratings for synthesized samples that surpass human reference quality. The second variant reduces ambiguity, as indicated by the relatively lower variance across raters. By combining these approaches, we achieve both more reliable and more fine-grained assessments. We also release MANGO[1], a massive dataset of 246,000 human ratings, the first-of-its-kind collection for Indian languages, aiding in analyzing human preferences and developing automatic metrics for evaluating TTS systems .

## 1 Introduction

Human evaluation is widely regarded as the gold standard for Text-To-Speech (TTS) assessment; however, it lacks standardization. This issue is more realized with the rapid advancements in TTS synthesis, where numerous models claim superiority over prior systems or human speech (Li et al., 2023; Wang et al., 2023; Tan et al., 2024). Deciphering the true extent of improvement from one model to the next is highly challenging due to inconsistent and often inadequately described subjective evaluation methodologies across studies.

The above problem is well studied for the Mean Opinion Scores (MOS) test (Wester et al., 2015; Finkelstein et al., 2023; Kirkland et al., 2023; Le Maguer et al., 2024) which has received much constructive criticism over the past few years. Specifically, in a MOS test, listeners assess each system independently, which can result in an inability to accurately capture the subtle relative differences between similar systems. This poses a significant challenge in modern TTS evaluation where systems that perform equally well need to be compared against each other. To address these issues some of the recent works rely on CMOS tests (Loizou, 2011). However, this test is costly and time-consuming as it involves $\binom{N}{2}$ comparisons between all pairs of $N$ systems.

---

[1]The dataset is publicly available at `https://huggingface.co/datasets/ai4bharat/MANGO`.

The MUSHRA test has been gaining popularity in addressing these issues. This test scales better by enabling a parallel comparison of the $N$ systems, and addresses the limitations of MOS tests that only allow isolated evaluation. However, we show that even the MUSHRA test is not devoid of issues. To begin with, we note that the MUSHRA test was conventionally designed to assess intermediate-quality audio systems (ITU-R, 2015). However, state-of-the-art TTS systems (Ju et al., 2024) are not of intermediate quality and instead generate audios having quality on par or even better than human recordings. To align with these modern developments, several works adopt variants of MUSHRA (Merritt et al., 2022; Li et al., 2023; Shen et al., 2024), which differ in implementation but the validity of these modified tests is unknown.

Given this situation, we critically assess the reliability, sensitivity, and validity of the MUSHRA tests by asking a series of research questions, such as: Is MUSHRA a reliable test, consistently yielding results comparable to other widely adopted subjective tests such as CMOS? Is the mean statistic reported in MUSHRA reliable, or is there significant variance across listeners and utterances? How sensitive is MUSHRA to implementation details? Particularly, how many listeners and utterances are required to yield statistically significant results? Is the conventional MUSHRA reject rule appropriate when TTS outputs sometimes outperform ground-truths? How does the choice of anchor affect MUSHRA scores, and what is the optimal anchor? While some of these questions have been studied for MOS (Wester et al., 2015), a comprehensive assessment of MUSHRA remains lacking.

With the goal of seeking answers to the above questions, we collected 246,000 human ratings by conducting the MUSHRA test involving 3 systems across two languages, viz., Tamil and Hindi. Our in-depth analysis based on these ratings, reveals two primary shortcomings: (i) reference-matching bias and (ii) judgement ambiguity. To mitigate these issues, we propose two refined variants of the MUSHRA test. The first variant does not explicitly identify the human reference to the rater. Doing so, prevents unfair penalties for well-synthesized samples that differ from the human reference, such as those with natural prosody that do not match the reference's prosody. In the second variant, raters are provided scoresheets to systematically calculate MUSHRA scores, by explicitly marking pronunciation mistakes, unnatural pauses, digital artifacts, word skips, liveliness, voice quality, rhythm, etc. Using the scores for these fine-grained criteria, they arrive at the final MUSHRA score. Our studies show that both these variants lead to a more reliable evaluation with the second variant also allowing for fine-grained fault isolation during evaluation. While MUSHRA-DG does require additional time to complete the tests, we believe that, given the limitations of the current MUSHRA setup, a slightly more time-intensive solution is justified to ensure the integrity and reliability of the evaluation process. We then show that a combination of these two approaches that leverages their individual strengths ensures both consistency and granularity. It allows modern TTS systems to be evaluated without being unfairly penalized for surpassing the reference in naturalness or prosody. The detailed scoring for pronunciation, prosody, and other factors provides actionable insights, and helps practitioners understand precisely where their systems excel and where improvements are needed. This combination creates a more balanced and sensitive evaluation framework, offering a clearer and more reliable assessment of TTS system performance.

In summary, our main contributions are:

1. A comprehensive assessment of the reliability, sensitivity, and validity of the MUSHRA test implementation in evaluating modern high-quality TTS systems.

2. Identification of two primary shortcomings of the MUSHRA test: (i) Reference-matching bias and (ii) Judgement Ambiguity.

3. Proposal of two variants of MUSHRA aimed at addressing these shortcomings.

4. Large-scale empirical validation of proposed variants resulting in MANGO, a dataset of 246,000 ratings from 492 listeners across Hindi and Tamil, examining three TTS systems.

## 2 Related Work

**Perceptual Judgments.** TTS evaluation primarily relies on naturalness, which, while lacking a universally agreed-upon definition (Shirali-Shahreza & Penn, 2023), generally refers to how closely synthesized speech

resembles human speech in perception. This concept aligns with standard evaluation practices and is typically assessed through subjective listening tests such as MOS (Łańcucki, 2021; Ren et al., 2021; Kim et al., 2021), CMOS (Shen et al., 2024; Tan et al., 2024; Li et al., 2023) or MUSHRA (Lajszczak et al., 2024), where listeners rate speech samples based on their perceived realism and overall fidelity.

**Critiques of TTS Evaluation.** Prior works mainly focused on a critique of MOS tests. Wester et al. (2015) analyze results from the Blizzard Challenge 2013 and highlight that an adequate number of listeners and utterances are needed to accurately identify significant differences. Clark et al. (2019) find that MOS tests are context-sensitive and yield different results when evaluating sentences in isolation as opposed to rating whole paragraphs. MOS tests are also known to show high variance in ratings (Finkelstein et al., 2023), subject to how raters are chosen. Kirkland et al. (2023) realize the importance of reporting scale labels, increments, and instructions, and show how these variables can affect scores. A recent study (Cooper & Yamagishi, 2023) highlights the presence of range-equalizing bias in MOS tests. Chiang et al. (2023) analyze over 80 papers, noting insufficient description of evaluation details and its impact on evaluation outcomes. Similarly, Le Maguer et al. (2024) highlight the need for better evaluation protocols.

**Emergence of Modern Tests.** Several variants of MUSHRA have been employed to overcome known shortcomings. To evaluate the robustness of TTS trained on imperfect transcripts, Fong et al. (2019), adopt the MUSHRA test without an anchor and also provide text transcripts during evaluation. Taylor & Richmond (2020) measure impact of morphology using a hidden natural reference, and utterances containing out-of-vocabulary words. Aggarwal et al. (2020) extend the MUSHRA test to also measure emotional strength of the synthesised speech. Merritt et al. (2022) adopt MUSHRA for evaluating speaker and accent similarity, by including both an upper-anchor and lower-anchor along with hidden reference. Li et al. (2023) adopt a variant of the MOS test, similar to MUSHRA, for testing naturalness and speaker similarity.

**Learnings from Human Evaluations in NLP** Freitag et al. (2021) highlighted the need for comprehensive, standardized evaluation frameworks like Multidimensional Quality Metrics for MT, which is crucial for TTS too. Ethayarajh & Jurafsky (2022) show that the average of Likert ratings (as followed in MOS tests in TTS) can be a biased estimate potentially leading to misleading rankings. Amidei et al. (2019) discuss how insufficient descriptions can make it difficult to interpret evaluation results. Howcroft & Rieser (2021) emphasize that current evaluations are inadequate for detecting subtle distinctions between systems; a problem we find recurring in TTS evaluations. Direct assessments have been popular in WMT evaluations (Barrault et al., 2020; Akhbardeh et al., 2021), however Knowles (2021) highlight several of its issues, a caution that carries over to human evaluations for TTS. They also advocate evaluating multiple systems on the same subset of documents, a practice we mirror in this work using audio samples instead.

## 3  MANGO: A Corpus of Human Ratings for Speech

We introduce a new dataset, MANGO: MUSHRA Assessment corpus using Native listeners and Guidelines to understand human Opinions at scale. It is a first-of-its-kind collection for any Indian language, comprising 246,000 human ratings of TTS systems and ground-truth human speech in both Hindi and Tamil, making it an expensive endeavour but one that we hope will contribute meaningfully to research in speech evaluation. Given the shortcomings of Mean Opinion Score (MOS) and Comparative Mean Opinion Score (CMOS) tests, our goal is to critically examine a promising alternative—the MUSHRA test—by conducting a large-scale evaluation involving multiple raters, systems, and languages. To do so, we adopt the standard MUSHRA test (ITU-R, 2015).

Raters evaluate multiple stimuli on each page, including an explicit (mentioned) reference that serves as a benchmark for high-quality speech, along with an anchor and implicit (hidden) reference to calibrate judgments. Each stimulus is rated on a continuous scale from 0 to 100, which is also discretized: 100-80 (Excellent), 80-60 (Good), 60-40 (Fair), 40-20 (Poor), and 20-0 (Bad). We describe our evaluation setup below, and provide the detailed instructions provided to participants in Appendix A.1.

### 3.1 MUSHRA-Based Annotation Framework

To evaluate TTS systems, we use the standard MUSHRA test ITU-R (2015), a methodology originally developed for codec evaluation and widely used in speech and audio quality assessment. MUSHRA is particularly suited for listeners with normal hearing and experience in critical listening, as it requires evaluating multiple stimuli side by side while referencing a high-quality benchmark. Raters assess multiple stimuli per page, including an explicit reference (a high-quality speech benchmark), an anchor (a lower-quality version for calibration), and an implicit (hidden) reference to normalize judgments. Each stimulus is rated on a continuous 0–100 scale, discretized into five categories: 100-80 (Excellent), 80-60 (Good), 60-40 (Fair), 40-20 (Poor), and 20-0 (Bad). To ensure reliable and consistent evaluations, MUSHRA follows established guidelines according to the standard. Listeners must complete pre-test training to familiarize themselves with impairments and test signals. Stimuli should be around 10 seconds long, with a maximum of 12 seconds, to minimize listener fatigue and enhance response stability. An assessor's responses are excluded if they rate the hidden reference below 90 in more than 15% of test items. Tests must use either headphones or loudspeakers, but not both within a session, ensuring consistency across participants. When listening conditions are technically and behaviorally controlled, data from as few as 20 subjects can yield statistically reliable conclusions. These measures help maintain the rigor and reproducibility of MUSHRA-based speech evaluations, making it a robust choice for large-scale TTS assessment.

**Comparison with Alternative Tests.** MUSHRA offers several advantages over Mean Opinion Score (MOS) and Comparative MOS (CMOS) for speech quality assessment. MOS, despite its widespread use, has been criticized for its poor reliability and inability to distinguish between similar-sounding systems due to its single-stimulus nature and the absence of an explicit reference. While CMOS provides pairwise comparisons, it is inherently limited to evaluating only two systems at a time, making large-scale assessments both expensive and difficult to scale. In contrast, MUSHRA enables simultaneous comparisons across multiple systems, allowing listeners to switch between samples and directly compare quality, which enhances sensitivity to subtle differences that MOS often fails to capture. The continuous 0–100 scale provides finer granularity than MOS's coarse 5-point rating, allowing for more nuanced judgments. Additionally, MUSHRA is more efficient than exhaustive pairwise CMOS tests, as it allows multiple systems to be evaluated in a single trial rather than requiring numerous pairwise comparisons. Another key strength is reference-based calibration, where listeners rate systems relative to a known high-quality reference and an anchor, with the aim of reducing scoring variability. Ribeiro et al. (2015) further confirm MUSHRA's advantages, demonstrating that it more effectively distinguishes between models compared to MOS, reinforcing its suitability for evaluating modern TTS systems.

**Evolution of MUSHRA.** Over time, researchers have refined the MUSHRA framework to better align with the evolving demands of speech synthesis evaluation. While the standard MUSHRA test remains widely used, several variations have emerged, modifying key aspects such as reference anchoring and test structure to address specific challenges in assessing TTS quality. For instance, BaseTTS (Lajszczak et al., 2024) follows the MUSHRA framework but omits anchors, suggesting that explicit anchoring may not always be necessary for reliable judgments. NaturalSpeech2 (Shen et al., 2024) further questions the necessity of strict reference matching by employing a MUSHRA-like CMOS setup without an explicit reference. Additionally, Expressive-MUSHRA has been introduced to evaluate expressiveness in speech synthesis (Huybrechts et al., 2021; Varadhan et al., 2024), highlighting MUSHRA's flexibility in capturing prosodic variation and expressive nuances. Beyond these structural modifications, studies have also examined MUSHRA's effectiveness and limitations. For example, Merritt et al. (2018) critique the traditional 3.5 kHz anchor, arguing that it may not be ideal for modern TTS, as distortions in neural TTS systems differ from those originally considered in codec evaluations. These adaptations reflect a growing recognition that MUSHRA, while valuable, may require refinements to remain optimally effective for contemporary TTS evaluation. Recognizing this need, we systematically explore alternative test designs and their impact on evaluation reliability.

### 3.2 Online Annotation Platform

We enhance the webMUSHRA (Schoeffler et al., 2018) platform to address its key limitations. Specifically, we modify a fork (Pauwels et al., 2021) and introduce session management to enable saving test progress,

Table 1: Dataset statistics of MANGO.

| Language | # Ratings | Gender | | Age | | | | | # Participants in MUSHRA Variants | | | |
|---|---|---|---|---|---|---|---|---|---|---|---|---|
| | | Female | Male | 18-25 | 25-30 | 30-35 | 35-40 | 40+ | Original | NMR | DG | DG-NMR |
| Hindi | 127,500 | 73 | 163 | 140 | 60 | 18 | 11 | 7 | 113 | 102 | 20 | 20 |
| Tamil | 118,500 | 154 | 81 | 82 | 73 | 36 | 28 | 16 | 100 | 97 | 20 | 20 |

and thereby allowing for breaks for listeners. We integrate consent forms and controls, such as ensuring raters listen to all audio samples in their entirety, to ensure more reliable ratings. We also integrate an event-tracking system to analyze the time spent per page.

### 3.3 Synthesizing Speech Samples for Annotation

To generate samples for TTS evaluation, we train TTS systems on the Hindi and Tamil subsets of the IndicTTS database (Baby et al., 2016). Each language consists of recordings from a female and male speaker (Hindi: 20.17 hours; Tamil: 20.59 hours). We train FastSpeech2 (FS2) (Ren et al., 2021) with HiFiGAN v1 (Kong et al., 2020) and VITS (Kim et al., 2021) from scratch on the train-test splits using hyper-parameters suggested in a recent study (Kumar et al., 2023b). We finetune StyleTTS2 (ST2) (Li et al., 2023) from the LibriTTS checkpoint.

### 3.4 Annotation Process and Dataset Statistics

To ensure reliable evaluation, we recruited native speakers of the target languages through reputable recruitment agencies. These agencies played a vital role in guaranteeing participant demographics aligned with the target language of each test. Please refer to Section A.9 for details on recruitment, consent, and compensation. Once recruited, the annotators underwent a comprehensive training process comprising multiple sessions aimed at familiarizing them with the evaluation platform, test interface, and evaluation criteria. As part of this process, demo calls were conducted, where participants were introduced to the rating interface and guided through a sample rating task. These sessions allowed participants to ask questions and receive clarifications in real time. Additionally, a structured review process was implemented, wherein participants initially rated five pilot samples. This phase allowed them to seek clarifications, provide feedback, and ensure their understanding of the evaluation criteria before proceeding to rate the 100 test samples. All instructions provided to participants are included in Appendix A.6 for transparency. This approach not only improved their confidence, but also helped standardize the assessment process across all evaluators.

With the above process, we collected 246,000 human ratings for TTS systems. We also collect demographic information from a majority of the 492 participants, while a subset of 21 individuals preferred not to disclose their details. Table 1 shows the demographic distribution, the number of participants, and the total number of ratings across all MUSHRA tests and our proposed variants (MUSHRA-NMR, MUSHRA-DG, MUSHRA-DG-NMR) which are described in Section 5. The table reflects responses from participants who disclosed demographic details, while a small subset preferred not to share this information.

## 4 Key Insights on MUSHRA

In this section, we address the research questions outlined in Section 1 and identify key challenges based on ratings collected in the MANGO dataset.

### 4.1 Is MUSHRA A Reliable Test?

In Table 2, we present the results of the MUSHRA test among 3 systems and find VITS and ST2 score highest in Hindi and Tamil respectively. Surprisingly, all systems attain scores in the "Good" bin with MUSHRA scores between 60 and 80, while the reference surpasses all systems with scores in the "Excellent" bin. Given that state-of-the-art TTS systems are able to reach quality on par with references, one would expect a much smaller gap between the reference and systems. To confirm this, we conduct the more reliable

Table 2: MUSHRA scores for Hindi and Tamil using Anchor-X and Anchor-Y, respectively, as anchors (ANC). $\mu$ represents the mean, $\sigma$ represents the standard deviation, and the 95% confidence intervals (CI) are provided.

| System | Hindi | | | Tamil | | |
|--------|-------|-------|------|-------|-------|------|
| | $\mu$ | $\sigma$ | CI | $\mu$ | $\sigma$ | CI |
| FS2 | 64.17 | 22.89 | 0.42 | 64.98 | 19.23 | 0.38 |
| ST2 | 66.74 | 21.65 | 0.40 | 71.38 | 18.31 | 0.33 |
| VITS | 67.65 | 20.58 | 0.38 | 65.66 | 18.91 | 0.37 |
| ANC | 70.81 | 20.92 | 0.39 | 20.08 | 16.69 | 0.38 |
| REF | 84.18 | 15.49 | 0.29 | 85.22 | 15.98 | 0.31 |

Table 3: Mean Comparitive-Mean-Opinion-Scores (CMOS) with 95% confidence intervals for Hindi & Tamil.

| System | Hindi | Tamil |
|--------|-------|-------|
| REF | - | - |
| ST2 | -0.11 $\pm$ 0.08 | 0.24 $\pm$ 0.09 |
| VITS | -0.10 $\pm$ 0.07 | -0.57 $\pm$ 0.09 |
| FS2 | -0.66 $\pm$ 0.08 | -0.60 $\pm$ 0.09 |

but expensive CMOS test with 15 listeners in each language. In this test, we ask the rater to compare a given system, such as VITS, with a reference audio sample. The rater evaluates both the reference and the output from a system being tested without prior knowledge of which audio sample corresponds to which system, ensuring an unbiased comparison. The raters assign a single score ranging from -3 to +3 in increments of 0.5. A score of -3 indicates that System A is much worse than System B . A score of +3 indicates that System A is much better than System B. A score of 0 means that both systems are equal in quality. As seen from the scores in Table 3, CMOS indicates that the outputs synthesised by VITS and ST2 are very close in quality to the reference in Hindi and Tamil respectively, while MUSHRA scores do not reflect this at all. We hypothesize that listeners in the MUSHRA test are subject to various biases, one of which we term the *reference-matching bias*. This bias may lead to situations where systems that perform comparably to or better than the reference are rated less favorably, as listeners tend to focus on aligning their ratings with the reference outputs while evaluating the systems. While this may have been acceptable when TTS systems lagged behind human speech quality, it is undesirable in the current scenario where modern TTS systems often exceed the reference in aspects like naturalness and prosody (Li et al., 2023; Shen et al., 2024). This suggests that the MUSHRA test, in its conventional form, may no longer be sufficient for evaluating state-of-the-art TTS systems. Instead, alternative methodologies, such as the variants we propose in Section 5, may help ensure more fair and accurate assessments.

Further evidence questioning the role of the reference as the highest standard in MUSHRA evaluations is provided in Appendix A.2, where we analyze CMOS-derived preference ratings and highlight cases where TTS systems are notably preferred over the human reference.

## 4.2 How reliable is the mean statistic in MUSHRA scores?

As mentioned earlier, each rater rates 100 utterances. In Figure 1, we use box-plots to visualize the distribution of MUSHRA scores (y-axis) for each rater (x-axis) across these utterances for each system, including the reference and anchor. While we acknowledge that the figure may appear overwhelming, we believe it is crucial for conveying the comprehensive view across both raters and utterances. We make two important observations from the figure. First, many the individual box-plots (especially of a system) have a high variance indicating that the same rater rates the system very differently across utterances. Second, looking at the means of the box-plots across different raters, we observe that there is a high variance in the means, indicating ambiguity in the perception of the MUSHRA labels across raters. We refer to this phenomenon as *judgement ambiguity*. This highlights the shortcomings of reporting mean statistics for MUSHRA scores, even when reported with confidence intervals (CI).

To delve deeper into *judgment ambiguity*, we examine variations between two systems. We consider an utterance where the mean scores for the samples generated by VITS and ST2 are nearly identical, but the variance across raters for each system is high. This high variance indicates significant ambiguity. Upon listening to many such utterances and speaking to many raters, we hypothesize that the ambiguity likely stems from different raters focusing on different aspects of the generated samples. For instance, some raters

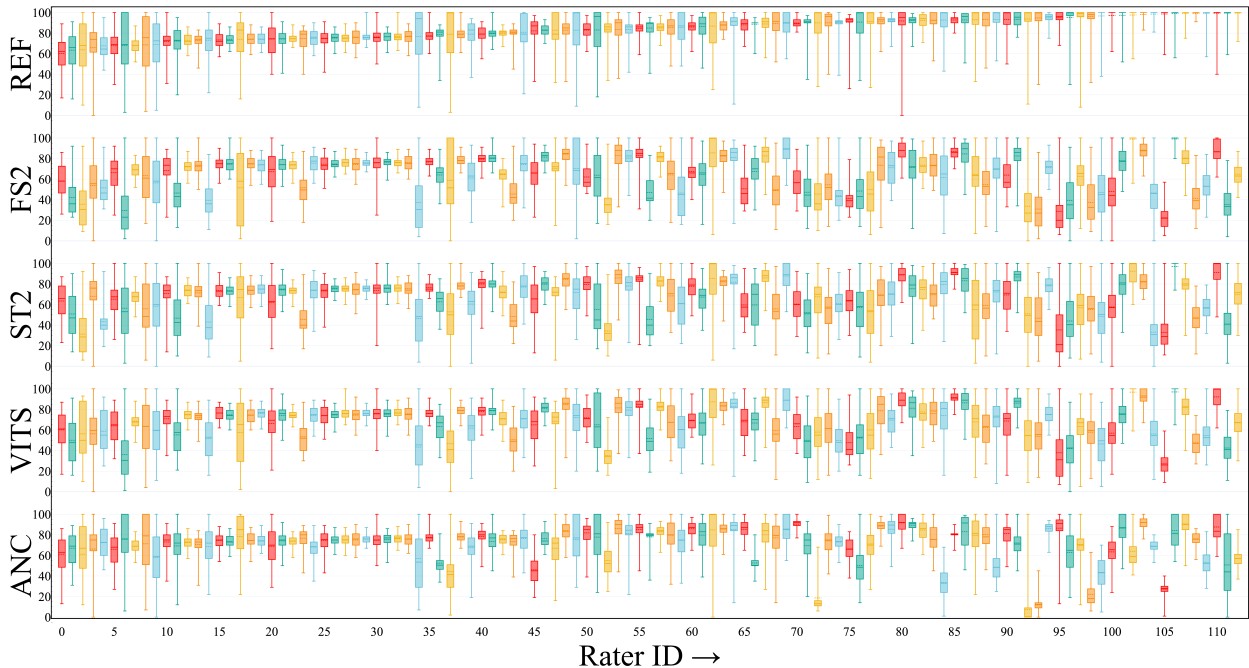

Figure 1: Visualization of the MUSHRA score distributions per rater across three systems— FS2, ST2, and VITS, along with the reference (REF) and anchor (ANC) for Hindi. Each boxplot represents ratings (0-100) across all test utterances for a system by one rater.While some boxplots exhibit relatively low variance, several display substantial height, indicating that certain raters assign widely varying scores to the same system across utterances. The variation in the means of the boxplot across raters suggests a high level of inter-rater variance. Raters are sorted in ascending order of their mean scores for the reference.

may prioritize prosody, others voice quality, and yet others the presence of digital artifacts. We hypothesize that asking raters to highlight these subtle differences across multiple dimensions while assigning a single score can lead to ambiguity in determining how much to penalize or reward a system's output. Hence, clear guidelines which take into account a fine-grained evaluation across different aspects would help (as proposed later in Section 5).

### 4.3   How sensitive is MUSHRA to number of listeners and utterances?

We use the procedure outlined in (Wester et al., 2015) to study the effect of number of listeners and utterances on MUSHRA scores. Specifically, we are interested in knowing if a smaller number of listeners and utterances would result in the same rankings of systems as obtained using the full set of listeners and utterances. To achieve this, we randomly sample a smaller subset of utterances and listeners and compute the mean system scores. We then calculate the Spearman rank correlation with the mean system scores obtained using all utterances and listeners. We repeat this process 1000 times, and compute the average over these large number of trials.

Figure 2 illustrates the average correlation of MUSHRA ratings in Hindi between a subset of listeners and utterances compared to the fully-scaled test (involving all listeners and utterances). Firstly, it is evident that using a minimum of 20 listeners is crucial to achieve correlations above 90%. Secondly, when employing a smaller number of listeners and utterances (e.g., fewer than 40 in both cases), increasing the number of listeners proves to be more beneficial than increasing the number of utterances. Using more than 30 listeners and 30 utterances invariably yields correlations above 95%. We notice similar trends for Tamil, but with higher correlations achieved with lesser number of listeners and utterances (Appendix 8). These results highlight the sensitivity of MUSHRA evaluations to the number of listeners and utterances, emphasizing the importance of careful selection and scaling of these factors to ensure reliable and meaningful evaluations.

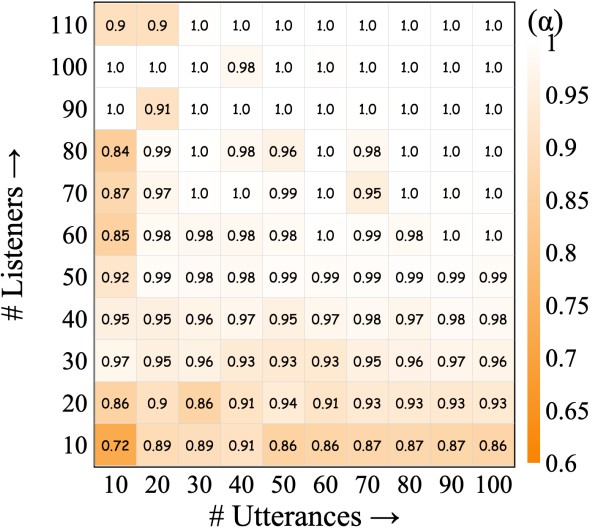

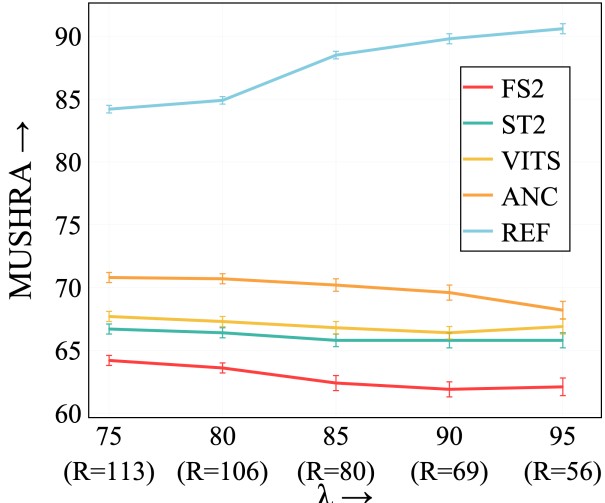

Figure 2: Rank correlation of mean scores obtained using subsets of listeners and utterances and mean scores obtained using all listeners and utterances in Hindi.

Figure 3: MUSHRA Scores in Hindi show score-variance but rank-invariance across systems when raters who rate Reference $\leq \lambda$ for more than 15% of utterances are rejected. R is the number of raters retained.

### 4.4 What is the impact of rejecting raters per standard MUSHRA protocol?

Traditionally, MUSHRA employs a rater rejection criterion, wherein raters scoring the hidden reference (HR) below a threshold ($\lambda$) more than 15% of the time are rejected. This rejection rule stems from the inherent assumption that the HR is the gold standard, which is not true in modern TTS settings where TTS systems (Li et al., 2023; Shen et al., 2024) are either on par with or surpass the reference. In such cases, a rater consistently scoring the HR lower might not necessarily reflect unreliability, but rather a nuanced perception of the reference's limitations compared to the evaluated systems. This is reinforced by observations from Table 3 where raters clearly prefer ST2 over reference for Tamil with a mean CMOS score of 0.24. This observation is further supported by the MUSHRA scores shown in Figure 3. The table shows that while rejecting raters based on the conventional MUSHRA criterion does not affect system rankings, it does notably shift scores. Specifically, system scores decrease with increasing $\lambda$, while reference scores increase. This trend hints at the *reference-matching bias* wherein raters who give the HR high scores might be unconsciously matching system samples to the mentioned reference, rather than rating based on their absolute perception of quality.

### 4.5 How does Anchor affect scores?

In MUSHRA tests, the anchor serves the purpose of setting the expectation of what a "Fair" sample sounds like. Typically, the anchor is created by minimally degrading the ground-truth by first downsampling it to 3.5 kHz and then upsampling it to 24 kHz. We refer to such an anchor as Anchor-X. In Table 2, the mean scores for Anchor-X in Hindi indicate that this anchor performs significantly better than all other systems, attaining a high score of 70.81. We believe these scores are explained by the resampling strategy used to create the Anchor-X, which introduces some artifacts in the audio but retains similar naturalness to the reference, especially in terms of prosody. Once again, this intuition indicates the tendency of raters to rate systems that match the reference with higher scores (reference-matching bias). We conclude that using this anchor may not be ideal, as it can lead to potentially "Excellent" TTS systems being unfairly rated as "Good". Essentially, the raters may perceive that if one of systems (anchor, in this case) sounds very similar to the reference, there is little justification for rating other systems highly.

Table 4: MUSHRA-Extended scores with 95% CI for Hindi.

| System | $\mu$ | $\sigma$ | CI |
|--------|-------|----------|------|
| FS2 | 63.12 | 21.30 | 0.93 |
| ST2 | 65.15 | 21.76 | 0.95 |
| VITS-R | 68.47 | 19.70 | 0.86 |
| VITS | 68.99 | 19.67 | 0.86 |
| ANC | 73.62 | 19.56 | 0.79 |
| XTTS | **73.65** | 18.52 | 0.86 |
| REF | 76.39 | 18.05 | 0.81 |

Table 5: Objective evaluation of TTS systems in Hindi and Tamil, measuring intelligibility, speech quality, and signal distortion.

| System | Hindi | | | Tamil | | |
|--------|-------|------|--------|-------|------|--------|
|        | STOI | PESQ | SI-SDR | STOI | PESQ | SI-SDR |
| ANC | 0.964 | 3.39 | 27.53 | 0.995 | 3.65 | 22.29 |
| ST2 | 0.997 | 3.72 | 24.72 | 0.998 | 3.99 | 28.63 |
| REF | 0.997 | 3.93 | 27.76 | 0.998 | 4.08 | 30.25 |
| VITS | 0.998 | 4.11 | 30.17 | 0.997 | 3.98 | 25.63 |
| FS2 | 0.999 | 4.17 | 28.89 | 0.998 | 4.09 | 26.74 |

Next, we study the use of an alternative anchor (Anchor-Y) that we know would likely fall in the "Poor" or "Fair" category, given its construction process. Specifically, we construct Anchor-Y by degrading ST2 outputs by averaging the pitch, reducing the number of diffusion steps, slowing the audio by 1.2 times, and inducing mispronunciation via the input text, along with word skips and word repeats in 20% of the samples. To obtain average voice quality, we set the number of diffusion steps to 3 with $\alpha = \beta = 0.8$. As expected, from the Tamil MUSHRA scores in Table 2, this anchor does indeed score poorly with a mean of 20.08. Interestingly, we see that despite a very low-quality anchor, other systems are not rated very highly, and there is still a huge disparity between ST2 (71.38) and the Reference (85.22). Given that high-quality anchors unfairly bias raters against other systems, while low-quality anchors seem to have no effect on the ratings for other systems, we believe there is merit in conducting MUSHRA evaluations without anchors (Lajszczak et al., 2024), which also saves costs by reducing human effort.

### 4.6 Does adding more systems affect scores?

To better understand cognitive overload in the MUSHRA test, we scale up the number of systems to be rated by introducing one new competitive system - XTTSv2, and repeating an existing system - VITS (VITS-R) in the original Hindi MUSHRA test. We finetune XTTSv2 (CoquiAI, 2023) starting from the multilingual checkpoint with the hyper-parameters from their original implementations on the same splits described in Section 3.3. We call this test - MUSHRA-Extended. The results show that raters were highly consistent, with VITS and VITS-R receiving nearly identical scores (68.99 and 68.47, respectively), despite the randomized order. Introducing XTTS, which outperformed other systems with a score of 73.65, did not disrupt the relative ranking of the remaining systems, which remained consistent with the original MUSHRA test. Thus, there does not seem to be significant cognitive overload, as we still observe consistent results. Note that we study cognitive load using $n = 7$ systems, but it remains to be seen how large $n$ can be before cognitive overload starts impacting the scores. For detailed scores, please refer to Table 4.

### 4.7 Do objective metrics align with human perception?

To complement our subjective evaluations, we compute three widely used objective metrics — (i) PESQ (Perceptual Evaluation of Speech Quality), (ii) STOI (Short-Time Objective Intelligibility), and (iii) SI-SDR (Scale-Invariant Signal-to-Distortion Ratio), using TorchAudio-Squim (Kumar et al., 2023a). PESQ estimates speech quality by assessing distortions caused by noise and compression, while STOI measures intelligibility by evaluating how well speech information is preserved. SI-SDR, commonly used in speech enhancement, quantifies the level of distortion in a signal relative to a clean reference. These metrics offer quantifiable insights, but are primarily designed and validated on English datasets, making their applicability to languages like Hindi and Tamil uncertain.

Table 5 reveals weak alignment between MUSHRA scores and both PESQ and STOI, suggesting that these metrics do not fully capture perceptual speech quality. This misalignment may stem from their focus on intelligibility and signal distortions rather than higher-level perceptual attributes such as naturalness, prosody, and timbre. In contrast, SI-SDR exhibits a strong Pearson correlation of 0.76 with MUSHRA scores, indicat-

ing that the degree of signal distortion plays a key role in perceptual judgments, or that SI-SDR effectively captures aspects of TTS quality that influence human ratings. These findings highlight the limitations of existing objective metrics and the need for more refined evaluation methods that better align with perceptual judgments, either as complementary measures or as reliable alternatives to subjective ratings.

## 5  Rethinking MUSHRA

We summarize two issues identified in Section 4. First, *reference-matching bias* that arises when listeners rate systems that perform at or above the level of the reference lower than deserved due to their efforts to align system outputs with the reference during evaluation. Second, *judgement ambiguity* that arises when listeners rate a system on a single scale using broadly defined metrics like "naturalness", leaving room for subjective interpretation of sub-criteria such as "prosody", "voice quality", "liveliness", etc. leading to high variability in ratings. Findings from our post-evaluation survey (Appendix A.5) further confirm the presence of these challenges, highlighting the influence of reference-matching and the inconsistencies in how raters interpret evaluation criteria. In response to this, we propose two refined variants of the MUSHRA test to address the identified challenges, as described below.

**MUSHRA-NMR.** The first variant, MUSHRA-NMR (MUSHRA with No Mentioned Reference), aims to mitigate the reference-matching bias observed in our analysis. MUSHRA-NMR follows all other standard protocols of the MUSHRA (ITU-R, 2015) test, except for the omission of the explicitly mentioned ground-truth reference that is presented to the listener. In the absence of this explicilty mentioned reference, the listener will be able to independently assess the quality of the TTS systems without trying to match them against the reference.

**MUSHRA-DG.** The second variant, MUSHRA-DG (MUSHRA With Detailed Guidelines), introduces comprehensive guidelines to reduce the ambiguity in rating samples for naturalness. In this test, we present raters with scoresheets and a formula to arrive at MUSHRA scores systematically. Each rater was asked to mark the number of (i) mild pronunciation mistakes, (ii) severe pronunciation mistakes, (iii) unnatural pauses, speedups, or slowdowns, (iv) digital artifacts, (v) sudden energy fluctuations, and (vi) word skips. Further, raters were also asked to rate more perceptual measures such as (i) liveliness, (ii) voice quality, and (iii) rhythm on a continuous scale from 0-100. The detailed guidelines provided to raters to assess across each of these dimensions can be found in Appendix A.6. We analytically derive a MUSHRA naturalness score for raters using an intuitive formula with weights (provided in Appendix A.6) for different dimensions listed above. These weights can be tweaked depending on the specific use-case. For example, in a TTS application designed for audiobooks, where fluidity and expressiveness are crucial for user engagement, we might assign higher weights to liveliness and rhythm.

We understand that devising a scoring formula involves some subjectivity. To address this, we encouraged raters to review their evaluations and adjust their fine-grained scores if they felt the overall scores from the formula did not accurately reflect the differences they perceived between system pairs. Notably, we tracked these revisions and found that they occurred in only 1.3% of cases. This suggests that raters likely found the computed scores to align with their assessments. Such alignment is reasonable, as the test design and the formula, including its specific weights and penalties, were refined through iterative evaluations and sustained engagement with raters over several months. Throughout this process, adjustments to the guidelines were made based on direct feedback and practical insights from raters, while also being guided by the intuition of TTS practitioners.

More interestingly, we notice that the scores derived from the MUSHRA-DG test preserve the rankings obtained from the gold-standard Comparative Mean Opinion Score (CMOS) tests (Table 3), thus reinforcing the validity of our evaluations. Additionally, the variance in MUSHRA scores calculated using the formula is significantly lower, indicating reduced ambiguity in ratings and providing a clearer distinction between different systems' performances.

Table 6: Comparison of MUSHRA scores and Proposed Variants for Hindi and Tamil languages.

| Language | System | MUSHRA-NMR | | | MUSHRA-DG | | | MUSHRA-DG-NMR | | |
|----------|--------|-----|-----|--------|-----|-----|--------|-----|-----|--------|
| | | $\mu$ | $\sigma$ | 95% CI | $\mu$ | $\sigma$ | 95% CI | $\mu$ | $\sigma$ | 95% CI |
| Hindi | FS2 | 61.99 | 23.86 | 0.46 | 72.73 | 11.65 | 0.51 | 81.51 | 11.50 | 0.50 |
| | ST2 | 68.09 | 22.01 | 0.43 | 73.41 | 12.03 | 0.53 | 84.97 | 12.22 | 0.54 |
| | VITS | **68.75** | 21.04 | 0.41 | **75.62** | 10.97 | 0.48 | **85.68** | 10.29 | 0.45 |
| | Anchor-X | 71.83 | 19.97 | 0.39 | 80.67 | 13.57 | 0.59 | 88.45 | 7.38 | 0.32 |
| | Reference | 76.39 | 18.08 | 0.35 | 90.87 | 9.34 | 0.41 | 88.63 | 7.39 | 0.32 |
| Tamil | Anchor-Y | 21.94 | 16.74 | 0.38 | 45.00 | 11.47 | 0.35 | 54.66 | 17.91 | 0.46 |
| | FS2 | 66.77 | 19.12 | 0.35 | 82.36 | 8.06 | 0.32 | 82.87 | 10.48 | 0.35 |
| | VITS | 68.52 | 18.28 | 0.36 | 81.96 | 7.41 | 0.35 | 83.86 | 10.12 | 0.44 |
| | ST2 | **76.64** | 17.68 | 0.33 | **88.82** | 7.88 | 0.50 | **91.10** | 8.05 | 0.78 |
| | Reference | 78.69 | 17.26 | 0.34 | 94.61 | 6.98 | 0.31 | 95.99 | 6.86 | 0.30 |

# 6 Results

We present human evaluation results of our proposed MUSHRA variants from the MANGO dataset.

## 6.1 Evaluations using MUSHRA-NMR

**Does our proposed variant help mitigate the reference-matching bias?**

In Table 6, we present the results of the MUSHRA-NMR test. We find the results to be rank-consistent with the scaled-up MUSHRA tests (Table 2). In the case of Tamil, we observe that the best performing system (ST2) is now scored much closer to the reference, clearly suggesting that the reference-matching bias has been mitigated. We observed that the score assigned to the reference itself decreased, indicating that the raters were strict. In the case of Hindi, the gap between the best performing system and the reference has again decreased but is not as small as in the case of Tamil.

We want to re-emphasize that this expectation of a reduced gap between the system and reference scores is well-founded. Feedback from TTS practitioners, including some of the authors who are native speakers, revealed that while some of the systems performed impressively in practice, the original MUSHRA scores did not seem to fully reflect their quality. This shortcoming is also clearly seen by the significant score differences between the reference and the other systems in the original MUSHRA test, whereas the CMOS scores in Table 2 indicate a closer alignment of a system's performance with the human ground-truth reference. Collectively, our findings above reinforce the merits of our proposed MUSHRA-NMR variant, which offers more reliable relative assessments compared to the MUSHRA test while retaining the advantages MUSHRA has over the CMOS test.

**How sensitive is MUSHRA-NMR to number of listeners and utterances?**

Subjective evaluations are often resource-intensive, making it desirable to minimize the number of listeners and utterances without compromising assessment quality. To explore this, we present the correlation between the scores derived from the MUSHRA variants using a subset of listeners and the scores obtained from the complete listener set, as shown in Figure 4. We also do a similar comparison across utterances. Our findings reveal that MUSHRA-NMR achieves a Spearman rank correlation exceeding 95% with the fully scaled-up MUSHRA test using just 20 utterances or 40 listeners. This indicates that significant reductions in both parameters are possible while maintaining reliability. Interestingly, our analysis indicates that enhancing the number of listeners has a greater impact on the accuracy of assessments compared to simply increasing the number of utterances.

## 6.2 Evaluations using MUSHRA-DG

**Does our proposed variant help mitigate judgement ambiguity while rating?** In Table 6, we show the effects of presenting detailed guidelines along with scoresheets to 20 participants to systematically

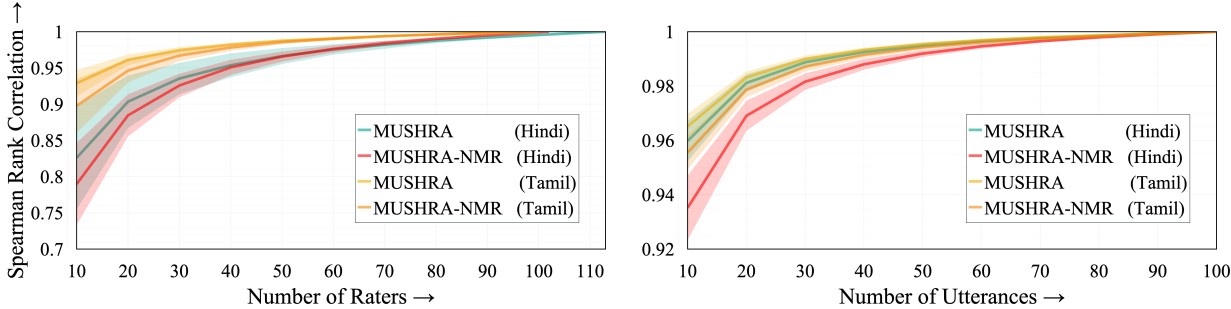

Figure 4: (Left) Correlation between scores from a subset of listeners and all listeners. (Right) Correlation between scores from a subset of utterances and all utterances.

arrive at MUSHRA scores. We find MUSHRA-DG scores to be rank-consistent with the CMOS tests , while systems scores are much higher and closer to the "Excellent" label, as expected. More importantly, the standard deviation of scores across all systems reduced by 43% in Hindi and 53% in Tamil when compared to the original MUSHRA, indicating that our proposed variant is able to reduce the ambiguity of rating naturalness on a single bar while preserving ranks.

It is important to explicitly address the apparent discrepancy in the ranking of VITS and FS2 systems in Tamil to clearly validate our claim that our proposed test also preserves rankings. While the MUSHRA-DG scores for these systems in Tamil suggest a reversed ranking between these systems, a closer inspection at the scores reveals that their perceived naturalness is highly comparable. This observation is also corroborated by the CMOS scores presented in Table 3, which show minimal separation between the two systems in Tamil. To confirm this more appropriately, we conducted a focused CMOS test that directly compared FS2 and VITS. Using a scale from +3 (indicating FS2 is significantly better) to -3 (indicating VITS is significantly better), 15 listeners rated the systems, and we obtained a CMOS score of 0.13. This result reinforces the notion that the two systems perform similarly.

While system rankings in closely matched scenarios can be debated, we argue that such cases highlight the need to focus on fine-grained differences. Understanding the specific contexts in which one model outperforms another provides valuable insights into system behavior and guides targeted improvements. Keeping this in mind, we subsequently discuss the effectiveness of the MUSHRA-DG test in fault isolation.

**Fault Isolation.** We collate the scoresheets of participants to obtain more fine-grained insights on where each model underperforms. In Figure 5a, we report the error rates of instances where an attribute received a rating greater than 0 for the six objective attributes and in Figure 5b the absolute perceptual scores on a scale of 100 for the remaining attributes. The granular ratings reveal the true power of this test in identifying defects in TTS outputs, especially among systems that achieved similar mean scores in the original MUSHRA. Specifically, we observe that for Hindi, a deterministic system like FS2 performs well in terms of pronunciation but suffers in prosody and word-skipping. Conversely, the close difference between VITS and ST2 is better explained by noting that VITS nearly outperforms in all dimensions, except that VITS exhibits nearly twice as many sudden energy fluctuations as ST2 and performs slightly worse in terms of rhythm.

Similarly, for Tamil, as detailed in Figure 10 in Appendix A.7, the marginal differences across dimensions clarify the perceived inconsistency in the rankings of VITS and FS2 mentioned earlier, which, upon closer inspection, is not truly an inconsistency but rather a reflection of their comparable overall performance. We also take this opportunity to address the inflated anchor scores observed in the MUSHRA-DG test compared to the original MUSHRA scores in Tamil. One explanation for this inflation could be attributed to leniency in the scoring formula applied to subjective dimensions, resulting in higher anchor scores. However, a more compelling explanation is that the original MUSHRA test may be influenced by a Range Equalizing Bias Zielinski (2016), where participants tend to stretch scores across the entire scale. Consequently, low-quality anchors are often relegated to the extreme lower end of the scale, even when the perceived degradation is

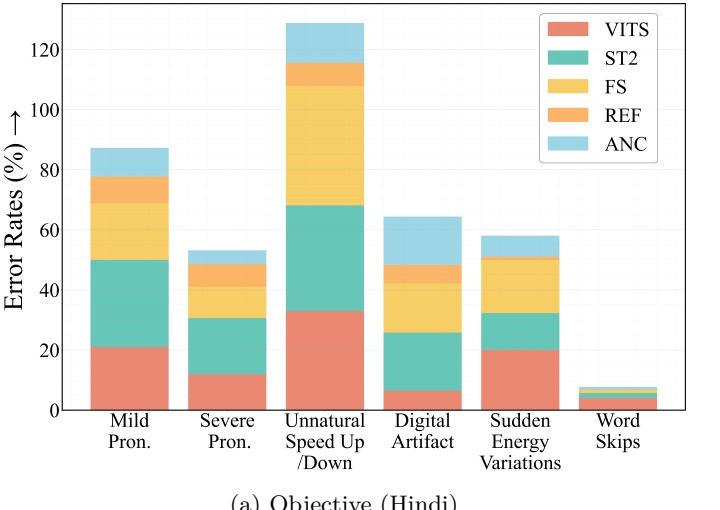
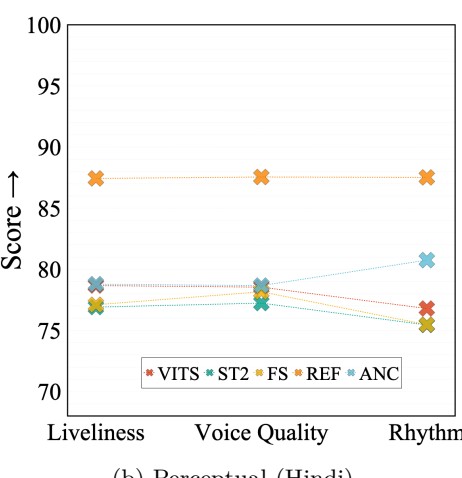

(a) Objective (Hindi)  (b) Perceptual (Hindi)

Figure 5: Visualization of the six objective and three perceptual dimensions of the MUSHRA-DG test. The objective scores are represented using stacked bars, where multiple error categories are displayed cumulatively rather than as independent percentages. The subjective dimensions are represented using a scatter plot with scores ranging from 0 to 100.

not as severe. In contrast, fine-grained evaluations like MUSHRA-DG emphasize detailed attributes and systematic scoring, which mitigate this bias. This approach reduces the tendency to artificially stretch scores, leading to higher and arguably more accurate anchor ratings. Crucially, this inflation does not affect the relative rankings of systems or the conclusions drawn from MUSHRA-DG. By minimizing variance and enhancing scoring consistency, MUSHRA-DG continues to demonstrate its robustness as a reliable and effective diagnostic and ranking tool for TTS evaluation.

Moreover, while the Reference scores appear higher in MUSHRA-DG, this increase is not merely a case of inflation. Notably, the scores of the systems also increase, resulting in all systems being classified within the "Excellent" category. This trend aligns with the expectations set by the CMOS scores shown in Table 3. Furthermore, MUSHRA-DG brings the scores of the top-performing models closer to the Reference scores, a pattern consistent with the results from CMOS evaluations. This alignment highlights the effectiveness of MUSHRA-DG in providing fine-grained assessments that capture subtle distinctions between systems while ensuring the overall scores accurately represent system performance.

**Time Complexity of MUSHRA-DG.** We hypothesize that the additional detail of evaluating each audio sample across multiple dimensions inevitably increases the time required for participants to complete the test. To verify this hypothesis, we visualize the average time taken across pages in Figure 6 and find that the MUSHRA-DG test indeed takes nearly twice as much time as the original MUSHRA test. However, we believe this extra time results in a much more comprehensive understanding of TTS system performance, and allows for fine-grained fault isolation, making the trade-off worthwhile.

### 6.3 Evaluations using MUSHRA-DG-NMR

In Table 6, we present the results of our combined variant, wherein we provide detailed guidelines (DG) and remove the mentioned reference (NMR). We observe that the majority of system scores now align more closely with the reference ratings and predominantly fall within the "Excellent" category, as anticipated from the CMOS tests. Moreover, compared to the MUSHRA-NMR test, the variance in scores has significantly diminished, indicating a marked reduction in rating ambiguity.

As TTS practitioners and native speakers of the language, we would like to emphasize that, despite the relative rankings being preserved in nearly all variants of the test, the combined variant is more reliable

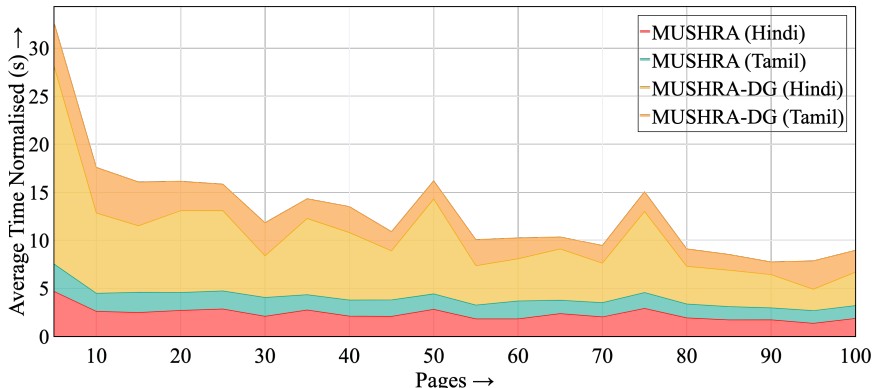

Figure 6: MUSHRA-DG exhibits higher average time (normalized by audio durations) across pages compared to MUSHRA.

because the scores now reflect the expected proximity of the systems to the reference (also established by the CMOS scores).

# 7 Generalization to English

Thus far, our study has primarily focused on Hindi and Tamil, two low-resource languages from distinct language families, Indo-Aryan and Dravidian, spoken by a combined 562 million people. This choice ensures coverage, diversity, and generalizability across language families in India. We do realize that evaluating our proposed MUSHRA variants on a high-resource language like English would further assess the broader applicability of our findings. Expanding the analysis requires additional resources, as human evaluations are both time and cost intensive. Given constraints, we focus our exploration on MUSHRA and the combined variant, MUSHRA-DG-NMR, drawing parallels with CMOS to analyze how our refinements translate to English. We conducted these evaluations on English TTS systems trained on the LJSpeech dataset, with 30 native US English speakers rating 30 utterances each, balanced across age (18 to 60) and gender. Additionally, we performed a CMOS test, where 15 participants compared all systems against the reference, rating 30 utterances each. Table 7 presents the results, which closely align with our earlier findings in Hindi and Tamil, reinforcing the robustness of our proposed evaluation methodologies.

## 7.1 Does the Reference-Matching Bias exist?

A key motivation for our work was to assess whether MUSHRA reliably captures the perceptual differences between TTS systems. The scores for English TTS systems in Table 7 reinforce our previous findings: while the reference (REF) is the only system placed in the "Excellent" bin, all other systems are restricted to the "Good" or "Fair" bins. However, CMOS results tell a different story, showing that ST2 and VITS are comparable to, or even preferred over, the reference. This discrepancy highlights MUSHRA's inability to fully capture perceptual parity or superiority when modern TTS systems approach or exceed reference quality.

These findings further confirm the presence of reference-matching bias (RMB) in English. As a result, MUSHRA scores may misrepresent true perceptual differences, reinforcing the need for evaluation methods that minimize the influence of reference dependence.

## 7.2 Does Judgement Ambiguity Exist?

To assess the reliability of the mean statistic in MUSHRA for English, we visualize the distribution of MUSHRA scores across raters and systems (see Appendix: Figure 13). These results reaffirm the two key issues previously observed in Tamil and Hindi -

Table 7: Comparison of MUSHRA, MUSHRA-DG-NMR, and CMOS for English.

| System | MUSHRA | | MUSHRA-DG-NMR | | CMOS | |
|--------|--------|--------|---------------|--------|--------|--------|
| | μ | 95% CI | μ | 95% CI | μ | 95% CI |
| ANC | 41.93 | 3.30 | 56.38 | 3.91 | - | - |
| FS2 | 56.86 | 3.76 | 63.33 | 3.32 | -0.78 | 0.22 |
| VITS | 72.77 | 2.75 | 75.89 | 2.96 | 0.02 | 0.12 |
| ST2 | 73.27 | 2.81 | 78.47 | 2.84 | **0.21** | 0.12 |
| REF | **82.62** | 1.54 | **79.57** | 2.96 | 0 | 0 |

1. **High variability in individual ratings:** A single rater does not always score a system consistently across different utterances, leading to fluctuations in their assessments.

2. **Significant perceptual differences across raters:** The mean scores assigned by different raters vary widely, indicating inconsistency in how MUSHRA labels are perceived and applied.

These findings highlight the presence of judgment ambiguity, where raters interpret and use the MUSHRA scale differently, leading to inconsistencies in scoring. This further reinforces our claim that the mean statistic alone is not a reliable measure of perceived quality, as it is strongly influenced by both intra-rater inconsistency and inter-rater variation.

### 7.3 Does our proposed variant (MUSHRA-DG-NMR) help?

Our proposed MUSHRA-DG-NMR test was designed to mitigate reference-matching bias (RMB) and judgment ambiguity, and the English results further confirm its effectiveness. As seen in Table 7, MUSHRA-DG-NMR brings system scores closer to the reference, suggesting that RMB is reduced. At the same time, it preserves the relative rankings of systems while shifting scores, mirroring our findings in Hindi and Tamil. This is particularly important, as it indicates that MUSHRA-DG-NMR does not distort system rankings but instead provides scores that may better reflect perceptual judgments.

While score variance in English does not show a significant reduction, a key strength of MUSHRA-DG-NMR lies in its ability to provide fine-grained insights beyond a single score. This is especially valuable when systems receive similar MUSHRA scores but differ in perceptual quality. By analyzing both subjective and objective dimensions, we can pinpoint the specific factors shaping listener preferences. For example, VITS exhibits nearly twice as many sudden energy variations as ST2, while ST2 has almost double the digital artifacts of VITS. Despite these trade-offs, ST2 is preferred overall due to its improved rhythmic consistency, reinforcing that a simple numerical score does not capture all aspects of TTS quality. For reference, visualizations of both objective and subjective dimensions are provided in Appendix A.7. Although MUSHRA-DG-NMR requires additional time for evaluation, its ability to uncover detailed system strengths and weaknesses makes it a valuable tool for TTS research and development. These fine-grained insights go beyond aggregated scores, offering a deeper understanding of where synthetic speech systems excel and where refinements are needed.

**Summary.** We release all English ratings, expanding our annotation dataset to 255,000 subjective annotations. These results confirm that MUSHRA scores are affected by reference-matching bias, mean scores can be unreliable due to rating inconsistencies, and MUSHRA-DG-NMR effectively mitigates bias while preserving system rankings. Additionally, our fine-grained analysis provides deeper insights beyond a single score, allowing for a more precise evaluation of system strengths and weaknesses. Overall, these findings further validate our conclusions across both high-resource (English) and low-resource (Hindi and Tamil) languages.

## 8 Future Work

While our study introduces refined MUSHRA variants to enhance the evaluation of modern TTS systems, several promising directions remain unexplored. The ITU-R standard ITU-R (2015) advises rating no more

than 12 signals at a time, yet real-world evaluations could require comparisons across a much larger set of systems for creating a leaderboard. A natural extension of our work would be to adapt MUSHRA-DG-NMR for tournament-style evaluations, allowing new systems to be seamlessly integrated while maintaining a dynamic leaderboard. Another compelling avenue is the development of objective metrics that can either complement subjective evaluations or, in certain cases, serve as viable replacements, offering a more holistic perspective on TTS quality while improving efficiency. Additionally, refining MUSHRA-DG to make it more accessible for large-scale studies could further extend its impact, without sacrificing the fine-grained insights it provides into pronunciation, prosody, and other key speech attributes. Beyond this, exploring cultural and demographic biases within the MUSHRA framework could illuminate potential disparities in how different listener groups perceive synthesized speech, leading to fairer and more representative evaluations. Furthermore, validating our refined MUSHRA variants in practical deployment settings will be crucial to ensure their seamless integration into real-world use cases, fostering broader adoption and iterative refinements based on real-world feedback. These directions build upon our proposed MUSHRA variants and pave the way for more rigorous, equitable, and insightful assessments of TTS systems.

## 9    Conclusion

Our comprehensive study reveals significant shortcomings in the current use of the MUSHRA test for evaluating modern high-quality TTS systems. Through an extensive analysis involving 246,000 human ratings for Hindi and Tamil, we identified two primary issues: reference-matching bias and judgment ambiguity. To address these issues, we propose two refined variants of the MUSHRA test: MUSHRA-NMR, which omits explicit identification of the human reference, and MUSHRA-DG, which uses detailed guidelines to calculate MUSHRA scores systematically. Our findings indicate that both variants lead to more reliable evaluations, with MUSHRA-DG offering the additional benefit of fine-grained fault isolation during assessment. Through this work, we also release MANGO, a large human rating dataset, to further support research in this area.

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

# A  Appendix

## A.1  Instructions For MUSHRA

**Instructions to Participants for MUSHRA Evaluations of Text-to-Speech Systems**

Thank you for participating in this speech evaluation study to assess the quality of various Text-to-Speech (TTS) systems. Please follow the instructions given below carefully.

**Overview**

In this evaluation, you will listen to different audio samples produced by various TTS systems. Your task is to rate these samples based on specific criteria using the MUSHRA (Multiple Stimuli with Hidden Reference and Anchor) methodology.

**Evaluation Procedure**

1. Listening Setup:

    a. Please use good-quality headphones or speakers to ensure you can hear all the nuances in the audio samples.

    b. Find a quiet space to minimise distractions during the evaluation.

    c. Use a consistent playback device throughout the evaluation to maintain uniformity in listening conditions.

2. Rating Scale:

    a. You will use a scale from 0 to 100 to rate the quality of each audio sample.

    b. The ratings correspond to the following categories:
        - 100-80: **Excellent**
        - 80-60: **Good**
        - 60-40: **Fair**
        - 40-20: **Poor**
        - 20-0: **Bad**

3. Listening and Rating:

    a. General Procedure: For each rating page in the MUSHRA test -
        I. Listen to the mentioned reference carefully to understand high quality.
        II. Then, listen to each system output. You can listen to samples multiple times if needed.
        III. Ensure you listen to each audio sample in its entirety without interruptions.
        IV. After listening to each sample, rate the quality of each of them based on its naturalness and overall quality.
        V. Please keep in mind that you can adjust your ratings as you listen to different samples.
        VI. Please take regular breaks after every 30 minutes to avoid strain and fatigue.

    b. Evaluation Criteria: After listening to each sample, rate the quality based on its naturalness and overall quality. Consider factors such as:
        - Naturalness: How similar does the audio sample sound to human speech?
        - Intelligibility: Is the speech clear and easy to understand?
        - Prosody: Does the output have appropriate intonation, rhythm, and stress?

    c. Comparative Assessment: Compare each sample with the others on the same page. Ensure that your ratings reflect the true relative rankings of the systems based on your perception. Your evaluations should capture the differences in quality as accurately as possible.

    d. Finalising Your Ratings:
        - Once you have rated all samples for a page, you may move to the next page.
        - Ensure that you are satisfied with your ratings before submitting, as they will be recorded.

If you have any questions or need assistance during the evaluation, please feel free to ask.

Figure 7: Guidelines sent to participants taking the MUSHRA test. They were given a live demo of the rating page and walked through the guideline sheet.

## A.2  CMOS Evidence of Limitations of Reference Matching

The assumption that the human reference always represents the highest quality standard in MUSHRA evaluations may not hold in practice, particularly as modern TTS systems continue to improve. Since MUSHRA inherently relies on reference matching, this approach may unintentionally penalize systems that perform on par with or even surpass the reference. If raters subconsciously anchor their judgments to the provided human reference, systems that differ from it, even in a perceptually superior way, may receive lower scores.

To investigate this, we analyze CMOS-derived preference ratings, where CMOS > 0 indicates a preference for the TTS system over the reference, CMOS < 0 indicates preference for the reference, and CMOS = 0 suggests no preference. As shown in Table 8, StyleTTS2 is preferred over the reference in 54% of cases in Tamil and 43.8% in English, demonstrating that model outputs can surpass human recordings in perceptual quality. Even VITS is preferred 37.2% of times against the reference in English, further challenging the assumption that the human reference should always be treated as the gold standard.

These findings provide direct evidence that MUSHRA's reliance on reference matching may skew evaluations when systems outperform the reference. This motivates us to explore MUSHRA-NMR, a variant where no

explicit reference is provided, allowing for fairer assessments when TTS models reach or exceed human quality. While inspired by multiple listening test methodologies, our goal remains to refine MUSHRA to better accommodate the evolving capabilities of TTS models while preserving its core strengths.

Table 8: CMOS-Derived Preference percentages for TTS Systems in English, Hindi, and Tamil.

| Language | System | Reference (%) | Equal (%) | System (%) |
|---|---|---|---|---|
| **English** | ST2 | 39.1 | 17.1 | **43.8** |
| | VITS | 38.7 | 24.1 | 37.2 |
| | FS2 | 68.3 | 13.5 | 18.2 |
| **Hindi** | ST2 | 42.8 | 18.0 | 39.2 |
| | VITS | 43.9 | 18.2 | 37.8 |
| | FS2 | 59.9 | 11.8 | 28.3 |
| **Tamil** | ST2 | 41.0 | 5.0 | **54.0** |
| | VITS | 63.6 | 4.8 | 31.5 |
| | FS2 | 62.9 | 3.5 | 33.5 |

## A.3 Visualizing MUSHRA Distributions

In Section 4, we discussed the distribution of MUSHRA scores across raters for Hindi using Figure 1. Similarly, in Figure 12 and Figure 13, we visualize the MUSHRA scores per rater across the systems for Tamil and English, respectively. Figure 14 and Figure 15, visualizes the MUSHRA scores for each utterance, averaged across raters, for Hindi and Tamil respectively.

## A.4 Sensitivity of MUSHRA in Tamil

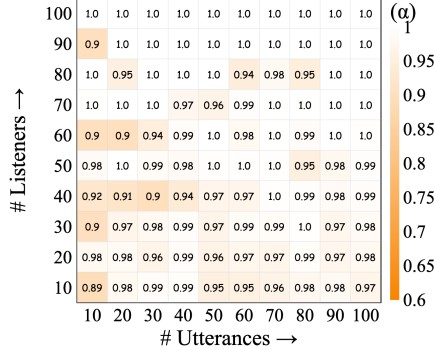

In Figure 8, we present the rank correlation of MUSHRA scores in Tamil, comparing a subset of listeners and utterances to the scores obtained using all listeners and utterances. Similar trends are observed as in the case of Hindi.

Figure 8: Spearman rank correlation of MUSHRA scores in Tamil

## A.5 Post-Evaluation Survey

To better understand how raters made their judgments during the MUSHRA test, we conducted a structured post-evaluation survey with 89 participants after the main evaluation. This survey provided direct insights into participant decision-making, particularly regarding reference-matching bias, judgment ambiguity, rating confusion, and cognitive load. The results, summarized in Figure 9, reinforce key concerns about strict reference dependence in MUSHRA-based evaluations.

A majority of participants (55%) preferred the system that was closer to the reference, confirming that reference-matching plays a strong role in decision-making. However, this bias was not absolute—18% preferred a system that deviated from the reference, and 11% rated solely based on overall quality, demonstrating

| Question | Hypothesis / Factor | Option | % |
|---|---|---|---|
| Please listen to the 3 audios below completely

1. Reference audio
2. System A
3. System B

Kindly mark your preference. | Reference-Matching | System B is better than System A because System B is closer to the Reference | 55 |
| | | System A is better than System B because System A is closer to the Reference | 20 |
| | | System B is better than System A in general. | 11 |
| | | System A is better than System B in general. | 7 |
| | | Can 't find the difference. | 7 |
| Please listen to the 3 audios below completely

1. Reference audio
2. System A
3. System B

Compare System A and System B against with the reference. System A made a pronunciation mistake at 2 words but had excellent voice quality. System B had average digital voice quality but showed excellent pronunciation with no mistakes. Which option would you choose? | Judgement Ambiguity | I would rate System A better than System B. | 56 |
| | | I would rate System B better than System A. | 33 |
| | | I would rate System A equal to System B. | 11 |
| How often did you come across confusing situations as the above? | Confounders | Only Few times did I face difficulty in assigning scores to different systems. | 56 |
| | | Always clear how to rate the different systems. | 31 |
| | | Many times faced difficulty in assigning scores to different systems due to ambiguity in scoring guidelines. | 7 |
| | | Every page I faced difficulty in assigning scores to different systems. | 6 |
| Please listen to the 3 reference audios below completely

1. Reference 1
2. Reference 2
3. Reference 3

On average, I felt the reference audio samples I listened to were - | Label Ambiguity [Reference] | Excellent (100 - 80) | 42 |
| | | Good (80 - 60) | 38 |
| | | Fair (60 -40) | 19 |
| | | Poor (40 - 20) | 1 |
| How would you rate below sample generated System A?

1. System A | Label Ambiguity [System] | Good (60 - 80) | 45 |
| | | Excellent (80 -100) | 35 |
| | | Fair (40 -60) | 12 |
| | | Poor (20 - 40) | 6 |
| | | Bad (0 - 20) | 2 |
| On a scale of 5, how difficult was the MUSHRA test? | Difficulty | Moderate | 40 |
| | | Easy | 34 |
| | | Difficult | 12 |
| | | Very easy | 10 |
| | | Very difficult. | 3 |

Figure 9: Summary of participant responses from the post-evaluation survey conducted after the MUSHRA test.

that strict reference dependence may not always be ideal. Furthermore, responses highlighted judgment ambiguity: while 56% prioritized voice quality over pronunciation errors, 33% favored better pronunciation over voice quality, showing that raters face trade-offs that are not always resolved by aligning with the reference.

Despite the provided instructions, 56% of participants reported experiencing rating confusion at least a few times, confirming that label ambiguity exists in the MUSHRA framework. Notably, 58% of raters did not rate the reference as "Excellent," with 19% rating it as Fair and 1% as Poor, further challenging the assumption that the reference always represents the highest quality standard. Additionally, 22% of participants found the MUSHRA test difficult or very difficult, suggesting that cognitive load may impact rating consistency.

These findings support our argument that while MUSHRA remains a valuable evaluation tool, strict reference-matching and scoring assumptions may not always be ideal. The observed ambiguity, cognitive challenges, and frequent encounters with confusing rating situations, reinforce the need for alternative evaluation setups, such as MUSHRA-NMR, which relaxes reference dependence while maintaining the benefits of comparative evaluation. The survey results directly informed our analysis, ensuring that our interpretations were grounded in participant responses rather than speculation. For additional details, the full survey findings are referenced in Figure 9.

## A.6 Detailed Guidelines for MUSHRA-DG

We present the complete guidelines shown to raters in Figure 16. We derive a formula that takes into account several factors: mild pronunciation mistakes ($MP$), severe pronunciation mistakes ($SP$), unnatural speedup or slowdown ($US$), liveliness ($L$), voice quality ($VQ$), rhythm ($R$), digital artifacts ($DA$), sudden energy fluctuations ($SEF$), and word skips ($WS$). The MUSHRA score is calculated by averaging the perceptual measures and then penalizing for various mistakes and artifacts. Specifically, we penalize every word skip by deducting 25 points, every severe pronunciation mistake by deducting 10 points, and every mild pronunciation mistake by deducting 5 points. Likewise, all other non-perceptual measures are penalized by 5 points. The MUSHRA score ($S_M$) for a system is given by,

$$S_M = \frac{L + VQ + R}{3} - \min(MP, 15) \times 5 - \min(SP, 7) \times 10 - US \times 5 - DA \times 5$$

$$-WS \times 25 - SEF \times 5$$

We believe this formulation ensures a systematic approach to scoring, accounting for both perceptual qualities and penalizing observable errors effectively.

## A.7 Fault Isolation in English and Tamil

Similar to Figure 5, we visualize the granular ratings across the objective and perceptual dimensions for English in Figure 11 and, Tamil in Figure 10.

## A.8 Limitations

Our study focuses on human evaluations for Hindi and Tamil, representing a major Indo-Aryan and Dravidian language, respectively. However, we did not extend our analysis to English, a widely spoken and diverse language. This limitation is due to the scope of our current research and resource constraints. We also wish to state that while the demographics table summarizes the age and gender of participants, details for 21 out of the 492 participants are missing as raters opted out from providing this information. This constitutes a minor fraction and we believe does not significantly affect the analysis. While this work focuses on identifying two key issues — reference-matching bias and judgment ambiguity — several other biases may also exist in TTS evaluations, which warrant further investigation. A more broader analysis could provide more comprehensive insights into the applicability and robustness of our proposed MUSHRA variants across diverse linguistic contexts and practical applications.

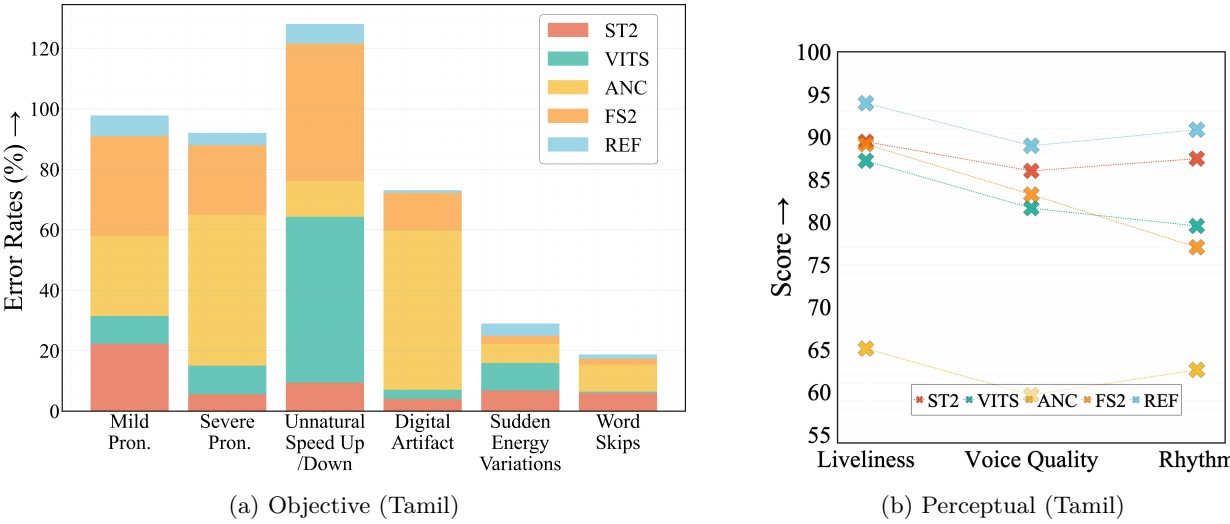

Figure 10: Visualization of the 6 objective and 3 perceptual dimensions of the MUSHRA-DG test.

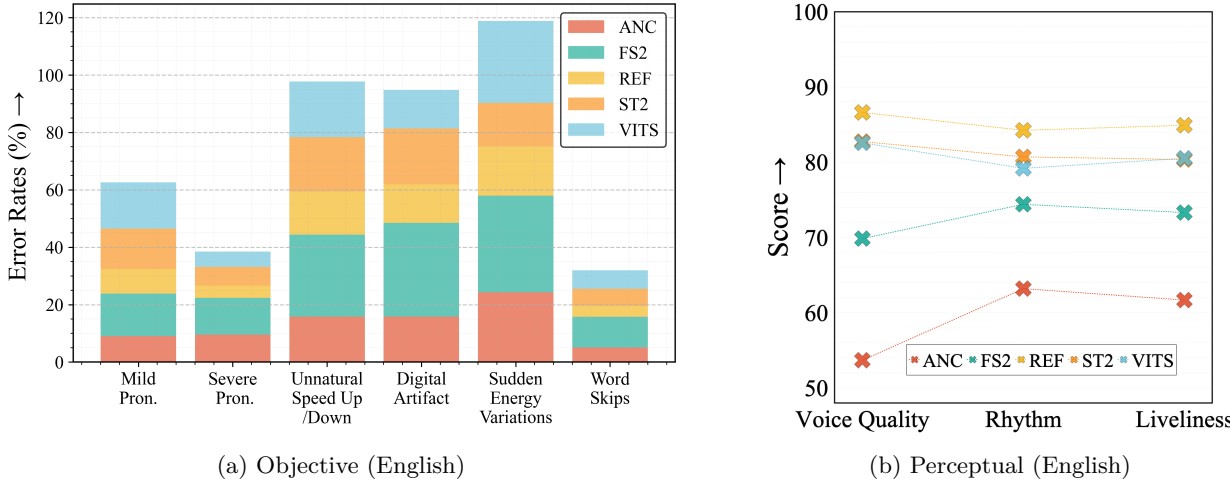

Figure 11: Visualization of the 6 objective and 3 perceptual dimensions of the MUSHRA-DG test.

## A.9    Ethical Considerations

We prioritized ethical conduct throughout our research. All human listeners involved in the study provided informed consent before participating in the evaluation, recruited through professional data annotation agencies. These agencies verified participant language proficiency for task relevance. We established an education criterion of completing high school to ensure participants' ability to accurately annotate audio content. Participants were compensated fairly for their time and expertise, following industry standards. They were also fully informed about the study nature, procedures, and their right to withdraw at any point without consequence. We ensured that our study adhered to ethical guidelines by obtaining approval from the Institutional Ethics Committee, which reviewed our methodology and confirmed compliance with ethical standards.

We strived for inclusivity and bias mitigation. Participants came from diverse demographic backgrounds, and we recruited only native speakers to capture the subtle linguistic and cultural nuances of each language. To minimize rater burden and bias in the new MUSHRA test variations, we prioritized user-friendliness and transparency in the design, providing clear guidelines.

We release the evaluations dataset under CC-BY-4.0 license after careful consideration of privacy and ethical use. Identifiable information about the participants was anonymized to protect their privacy. We encourage the use of this dataset for advancing TTS evaluation metrics, emphasizing that it should be used responsibly and ethically, adhering to principles of transparency and fairness. Finally, we acknowledge that our study focuses on Hindi and Tamil, and we recognize the importance of extending such evaluations to other languages, including English, to generalize our findings. Future research should continue to explore these ethical dimensions, ensuring that the development and evaluation of TTS systems are conducted with respect for the diversity and rights of all participants involved.

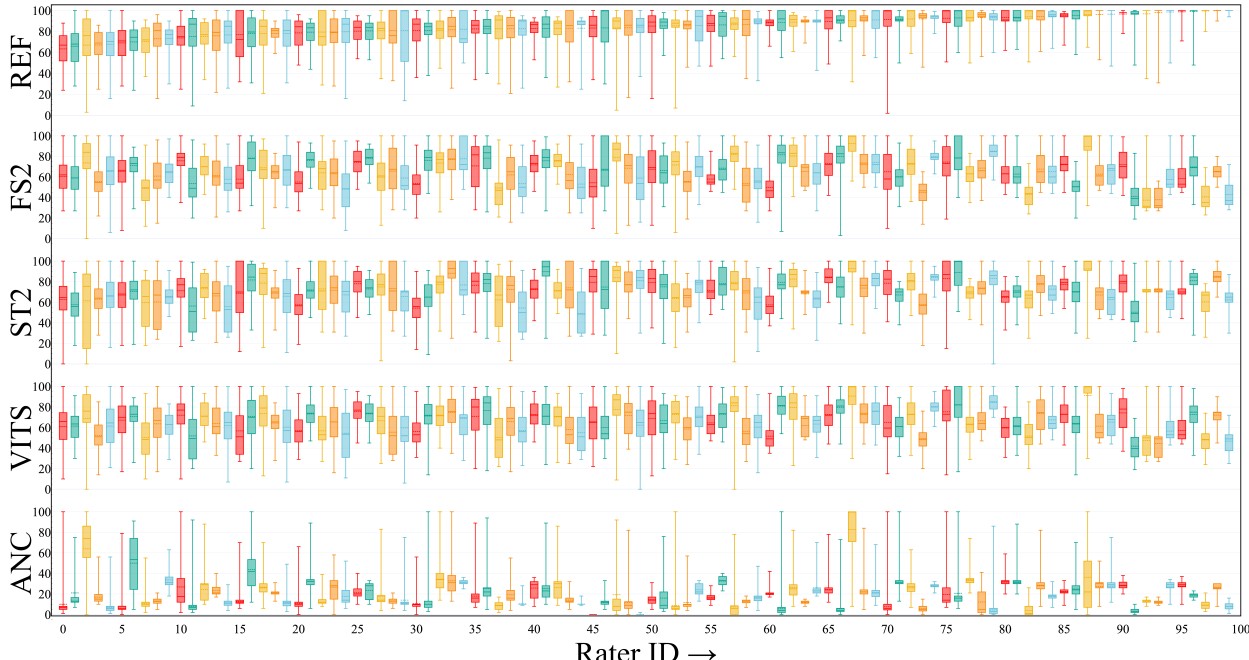

Figure 12: Visualization of the MUSHRA score distributions per rater across three systems— FS2, ST2, and VITS, along with the reference (REF) and anchor (ANC) for Tamil. Each boxplot represents ratings (0-100) across all test utterances for a system by one rater. The substantial heights of some boxplots indicate significant variance in the scores of that rater. The variation in boxplot means across raters suggests a high level of inter-rater variance

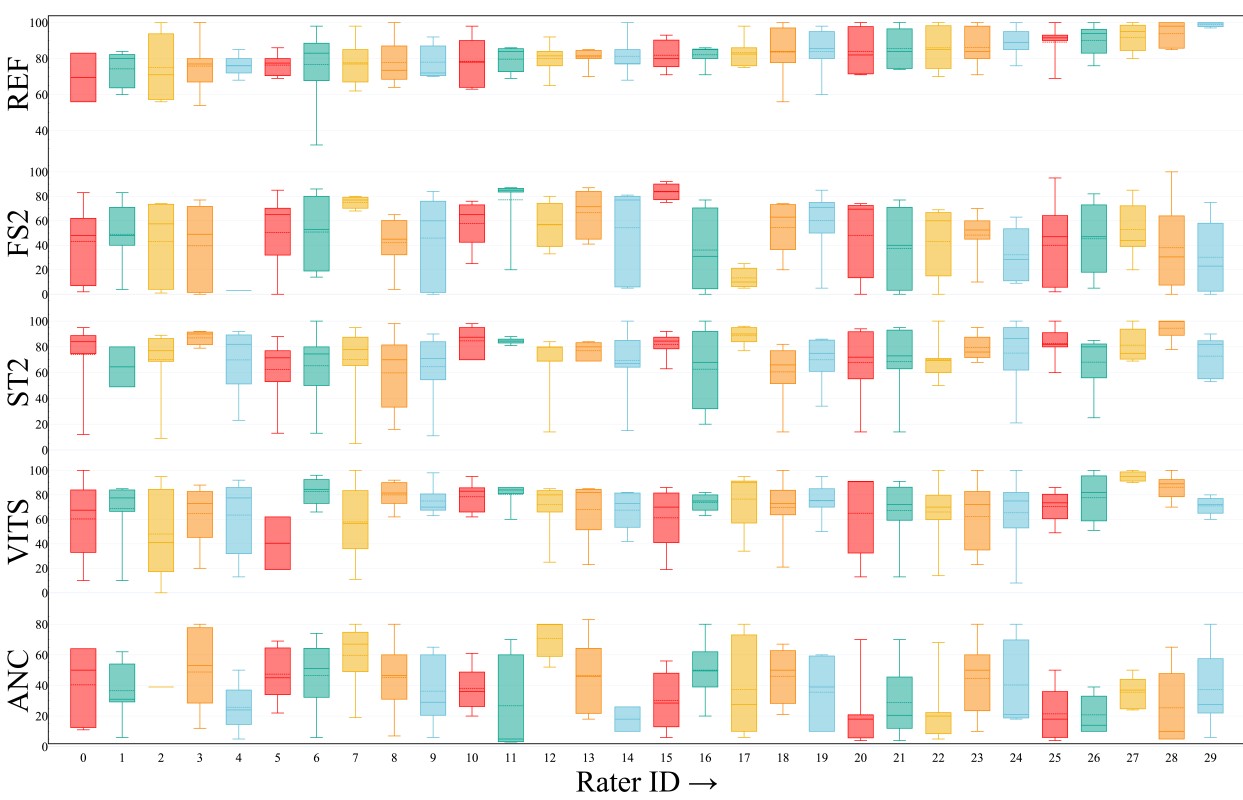

Figure 13: Visualization of the MUSHRA score distributions per rater across three systems— FS2, ST2, and VITS, along with the reference (REF) and anchor (ANC) for English. Each boxplot represents ratings (0-100) across all test utterances for a system by one rater. The substantial heights of some boxplots indicate significant variance in the scores of that rater. The variation in boxplot means across raters suggests a high level of inter-rater variance

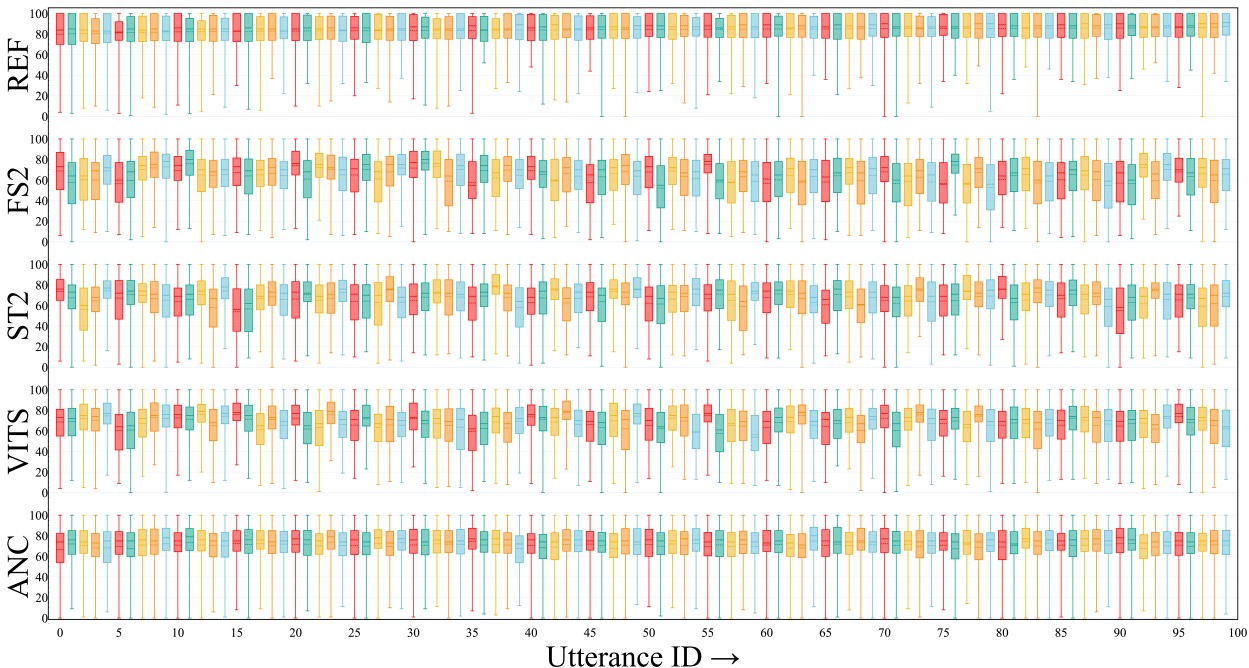

Figure 14: Visualization of the MUSHRA score distributions per utterance across three systems— FS2, ST2, and VITS, along with the reference (REF) and anchor (ANC) for Hindi. The X-axis represents each of the 100 utterances. Each boxplot represents ratings (0-100) across all raters for a system for a given utterance. The substantial heights of some boxplots indicate significant variance in the scores given by different raters for a single utterance.

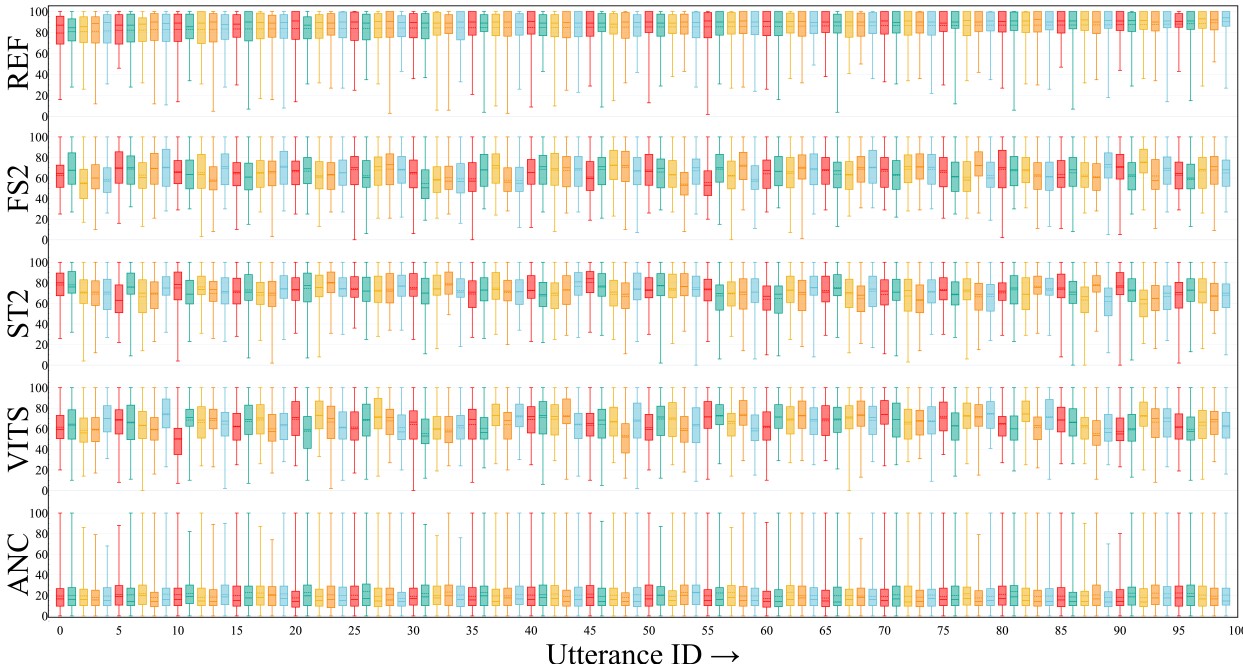

Figure 15: Visualization of the MUSHRA score distributions per utterance across three systems— FS2, ST2, and VITS, along with the reference (REF) and anchor (ANC) for Tamil.

| Criteria | To Mark |
|---|---|
| **Mild Pronunciation** | * Mark number of mild pronunciation errors.
* If no errors, mark 0 here.

A mild pronunciation error is where any character, for example an "r" or "t", is half-pronounced and not fully clear. |
| **Severe Pronunciation** | * Mark number of severe pronunciation errors. If no errors, mark 0 here.

A severe pronunciation error is where any character such as "r" or "t" is skipped/ mis-pronounced. |
| **Unnatural Pauses, speedup or slowdown** | * Mark number of places where there was unnatural pauses/speedup/slowdown in audio. |
| **Liveliness** | * Mark 100 if human-like
* Mark 85 if semi-expressive/ semi-enthusiastic/ semi-lively
* Mark 70 if robotic/monotonic

You may adjust scores in-between based on opinion. |
| **Voice Quality/Clarity** | * Mark 100 if perfect human like voice quality
* Mark 85 if slight digitalness in voice
* Mark 60-70 if high digitalness/persistent robotic voice

You may adjust scores in-between based on opinion |
| **Rhythm** | * Mark 100 if human-like
* Mark 85 if slightly fast/slow
* Mark 60 if too fast/slow

You may adjust scores in-between based on opinion |
| **Digital Artifacts** | * Mark number of digital artifacts heard in audio. If no artifacts, mark 0 here.

A digital artifact could be a "click" sound, "pop" sound, digital vibration in pauses, etc. |
| **Sudden Energy Fluctuations** | * Mark number of regions in which the energy, rhythm, pitch of the speech suddenly or irregularly change.

* Mark 0 here if no such changes noticed. |
| **Word Skips** | * Mark the number of words the model has skipped. If no skips, mark 0 here. |

Figure 16: Guidelines presented to raters across multiple evaluation criteria in the MUSHRA-DG Test.

