# OpenReview forum: "Rethinking MUSHRA: Addressing Modern Challenges in Text-to-Speech Evaluation"
_TMLR — Accepted by TMLR_

### Review · Reviewer_y228 · 2024-12-28

**Summary Of Contributions:**

This paper investigates the effectiveness and robustness of the MUSHRA subjective evaluation test for assessing Text-to-Speech (TTS) systems.

Specifically, the authors conduct experiments using two Indian language datasets: Hindi and Tamil, and evaluate three TTS systems: FastSpeech2, VITS and StyleTTS2. The MUSHRA tests are conducted with 492 local language listeners. Through these experimental investigations, the authors identify two key limitations of the MUSHRA tests: 1) evaluators tend to align their ratings with the explicit reference provided, and 2) using a single score for evaluation introduces ambiguity in assessing audio quality.

To address these two limitations, this paper propose two variants of MUSHRA: 1) MUSHRA with no explicitly mentioned reference, and 2) MUSHRA with detailed guidelines.

**Audience:**

Yes

**Broader Impact Concerns:**

No broader impact concerns.

**Claims And Evidence:**

Yes

**Requested Changes:**

The authors could also report objective audio quality evaluation metrics, such as SDR, PESQ, and ESTOI, for the tested TTS systems as a point of reference.

**Strengths And Weaknesses:**

Strengths:
- The paper presents thorough experiments and analysis that contribute to a deeper understanding of the MUSHRA test. The two limitations highlighted by the authors appear to be relevant and well-founded.
- The two solutions proposed by the authors seem to be effective. By removing the explicit reference, the gap between the best-performing TTS system and the hidden reference has been reduced. Besides, the use of more detailed scoresheets can help guide the evaluators in their assessments and decrease the variance in the evaluation scores. These detailed scoresheets also offer practitioners a better understanding of TTS system performance from different aspects (pronunciation mistakes, prosody, liveness, etc.)

Weaknesses:
- As the authors mentioned in Section 6.2, the MUSHRA test with detailed scoresheet is more time-consuming. Moreover, the formula used to compute the final score based on evaluations of different aspects (specified in appendix A.4) appears somewhat subjective and requires more rigorous justification.

---

> ### Author Response · Authors · 2025-01-30
> **Response to Reviewer y228**
>
> We sincerely thank Reviewer y228 for their thoughtful and constructive feedback. We appreciate the acknowledgment of the strengths of our work and the valuable suggestions for improvement.
>
>
> ### **Concern 1: Subjectivity of the Formula Used to Compute Final Scores**
>
> * We understand the reviewer's concern regarding the subjective nature of the formula used to compute final scores in MUSHRA tests with detailed scoresheets. To address this, we wish to clarify that we developed the scoring formula through multiple discussions with raters and researchers. These conversations helped us design a scoring method that aligns with how raters typically evaluate TTS systems, making it closer to real-world rating practices. The formula combines perceptual measures, such as liveliness, voice quality, and rhythm, with penalties for observable errors like pronunciation mistakes, unnatural speed changes, and digital artifacts.
>
> * The axes and weights in the formula were chosen based on feedback from these discussions. We observed that raters often focus on penalizing mistakes rather than explicitly rewarding correct features, which influenced how we designed the scoring process. We also considered the severity and impact of different types of errors during these conversations to create a formula that reflects what most raters prioritize, while allowing for some variation in individual preferences.
>
> * While we acknowledge that this methodology may benefit from further refinement in future iterations, it represents a systematic, empirically grounded, and practically validated framework. By balancing perceptual qualities and penalties for errors, we believe this formulation reflects the rigor of our approach and provides a reasonable and useful scoring mechanism. We hope this explanation addresses the reviewer's concerns.
>
>
> ### **Concern 2: Inclusion of Objective Metrics**
>
> We agree with the reviewer's suggestion to report objective metrics as complementary evidence. In response, we have computed and added the results of STOI, PESQ, and SI-SDR for all TTS systems. We find that these objective metrics only partially align with the subjective MUSHRA scores, particularly in cases where systems are closely comparable. The divergence may stem from the challenges of applying objective metrics, which are primarily designed for English, to TTS systems evaluated on Hindi and Tamil. Nonetheless, we appreciate the reviewer's suggestion to emphasize the importance of combining both subjective and objective evaluations for a more comprehensive assessment.
>
> Please find below the tables of computed objective metrics, which we have included in the revised manuscript.
>
> **Hindi**
>
> | **System** | **STOI** | **PESQ** | **SI-SDR** |
> |------------|:--------:|:-------:|:----------:|
> | ANC | 0.964 | 3.39 | 27.53 |
> | ST2 | 0.997 | 3.72 | 24.72 |
> | REF | 0.997 | 3.93 | 27.76 |
> | XTTS | 0.998 | 3.98 | 27.90 |
> | VITS | 0.998 | 4.11 | 30.17 |
> | FS2 | 0.999 | 4.17 | 28.89 |
>
> **Tamil**
>
> | **System** | **STOI** | **PESQ** | **SI-SDR** |
> |------------|:--------:|:-------:|:----------:|
> | ANC | 0.995 | 3.65 | 22.29 |
> | VITS | 0.997 | 3.98 | 25.63 |
> | FS2 | 0.998 | 4.09 | 26.74 |
> | ST2 | 0.998 | 3.99 | 28.63 |
> | REF | 0.998 | 4.08 | 30.25 |
>
> We sincerely thank the reviewer once again for their invaluable feedback, which has significantly contributed to improving the quality of our work. We hope these revisions adequately address the concerns. We will upload the revised manuscript once all the reviews are in. Please let us know if any further clarifications are required.
>
>
> Sincerely,
>
> Authors

---

### Review · Reviewer_1BMu · 2025-01-10

**Summary Of Contributions:**

This submission introduces significant advancements in evaluating Text-to-Speech (TTS) systems through a critical assessment and refinement of the MUSHRA (Multiple Stimuli with Hidden Reference and Anchor) test, a cornerstone in TTS evaluation. Below is a summary of the primary contributions:

#### 1. **Comprehensive Evaluation of MUSHRA**
   - Conducted a detailed analysis of the MUSHRA test using 246,000 human ratings across Hindi and Tamil, revealing critical shortcomings in the current setup:
     - **Reference-Matching Bias:** Listeners tend to rate systems lower if they deviate from the explicitly mentioned reference, even when surpassing the reference in quality.
     - **Judgment Ambiguity:** Ratings vary due to the lack of fine-grained evaluation criteria and subjective interpretations.

#### 2. **Proposed Variants of MUSHRA**
   - Introduced two refined methodologies to address the limitations:
     - **MUSHRA-NMR (No Mentioned Reference):** Eliminates explicit mention of the human reference to mitigate biases, enabling a fairer comparison of systems.
     - **MUSHRA-DG (Detailed Guidelines):** Provides systematic guidelines and a scoring formula to reduce ambiguity, introducing fine-grained criteria like pronunciation errors, rhythm, and liveliness.

#### 3. **Creation of the MANGO Dataset**
   - Released a novel dataset, **MANGO**, comprising 246,000 ratings for TTS evaluations, representing a diverse set of human preferences and linguistic nuances for Indian languages. This dataset enables further exploration of TTS evaluation methods and automatic metrics.

#### 4. **Empirical Validation and Insights**
   - Demonstrated that the proposed variants enhance reliability and granularity:
     - **MUSHRA-NMR** reduces the reference bias, yielding more accurate rankings.
     - **MUSHRA-DG** ensures lower variance in ratings and offers fault isolation capabilities.
   - Highlighted the sensitivity of MUSHRA to factors like listener count, utterance diversity, and anchor selection.

#### 5. **Actionable Framework for Future TTS Evaluations**
   - Proposed a combined variant, **MUSHRA-DG-NMR**, which synthesizes the strengths of both variants to provide a robust evaluation framework, ensuring fairness, reliability, and diagnostic utility.

#### 6. **Broad Impact**
   - Established the importance of adapting evaluation methodologies to account for advancements in TTS systems that can surpass human benchmarks, setting a precedent for fair and comprehensive assessments in AI and speech synthesis research.

**Audience:**

Yes

**Claims And Evidence:**

Yes

**Requested Changes:**

### Requested Changes

#### **Critical Changes**
1. **Generalizability Across Languages:**
   - Expand the analysis to include additional languages, particularly high-resource languages such as English, to validate the findings across diverse linguistic contexts. This is essential for ensuring the broader applicability of the proposed methodologies.

2. **Anchor Design Justification:**
   - Provide a more detailed rationale for the choice of anchors (Anchor-X and Anchor-Y) and discuss their potential impact on system scores. Additionally, consider exploring alternative anchor design strategies to mitigate any unintended biases.

3. **Scalability Discussion for MUSHRA-DG:**
   - Include a discussion on the time complexity of **MUSHRA-DG** and **MUSHRA-DG-NMR** in large-scale evaluation settings. Offer potential solutions to mitigate the increased evaluation time, such as parallelization or adaptive guidelines.

4. **Subjectivity in Scoring Formula:**
   - Justify the specific weights and penalties used in the **MUSHRA-DG** scoring formula or provide a framework for task-specific adaptations to improve its versatility.

#### **Suggested Enhancements**
1. **Bias Analysis:**
   - Investigate additional potential biases (e.g., cultural or demographic influences) in human evaluations to strengthen the study's fairness and reliability claims.

2. **Visualization Improvements:**
   - Enhance the clarity of the visualizations (e.g., box plots and fault isolation charts) by adding more descriptive legends and captions to improve interpretability for readers unfamiliar with the dataset.

3. **Dataset Accessibility:**
   - Provide more details about the **MANGO dataset**, including example ratings or summaries of specific entries, to illustrate its value for researchers better.

4. **Practical Applicability:**
   - Include case studies or examples of how the proposed variants (e.g., **MUSHRA-DG** and **MUSHRA-NMR**) have successfully applied in real-world TTS evaluations.

5. **Comparative Analysis with Other Tests:**
   - Deepen the comparison of the proposed variants with other evaluation methodologies, such as MOS and CMOS, to highlight relative strengths and limitations in various scenarios.

6. **Future Research Directions:**
   - Add a dedicated section on potential future research, such as adapting **MUSHRA-DG** for real-time applications or incorporating automatic metrics to complement human evaluations.

**Strengths And Weaknesses:**

### Strengths

1. **Comprehensive Analysis of MUSHRA:**
   - The submission offers an in-depth assessment of the MUSHRA test, addressing its limitations with empirical evidence derived from an extensive dataset (246,000 human ratings).
   - The authors critically evaluate key challenges such as reference-matching bias and judgment ambiguity, which are highly relevant to modern TTS evaluation.

2. **Innovative Variants Proposed:**
   - The introduction of **MUSHRA-NMR** and **MUSHRA-DG** is novel and well-motivated:
     - **MUSHRA-NMR** mitigates biases by removing explicit references.
     - **MUSHRA-DG** introduces fine-grained evaluation criteria, providing a more systematic and diagnostic approach to rating.

3. **Robust Dataset Contribution:**
   - The **MANGO dataset** significantly contributes to the community, representing diverse linguistic contexts (Hindi and Tamil) and offering a resource for benchmarking and studying human evaluations in TTS.

4. **Empirical Validation:**
   - The submission provides thorough empirical support for the proposed methods, demonstrating improved reliability, reduced variance, and enhanced diagnostic insights through fault isolation.
   - Results are well-illustrated using statistical correlations, variance analyses, and comparative studies with CMOS scores.

5. **Practical Relevance:**
   - The work addresses challenges faced by practitioners evaluating state-of-the-art TTS systems, aligning evaluation metrics with the realities of modern speech synthesis quality.

6. **Clear and Structured Presentation:**
   - The paper is well-structured, with clear motivation, methodology, and results. The inclusion of detailed guidelines for raters and a formula-based scoring approach in **MUSHRA-DG** adds practical value.

### Weaknesses

1. **Limited Language Scope:**
   - The focus on Hindi and Tamil is valuable but narrows the generalizability of the findings. Including results for widely-used languages like English could strengthen the study’s global applicability.

2. **Time Complexity of MUSHRA-DG:**
   - While the added granularity in **MUSHRA-DG** is a strength, the increased time required for evaluations may pose scalability concerns, particularly for large-scale studies.

3. **Subjectivity in Scoring Formula:**
   - The weights and penalties in the **MUSHRA-DG** scoring formula are subject to interpretation and may not align perfectly with all TTS use cases. More justification or adaptability of these parameters could improve its applicability.

4. **Anchor Design Limitations:**
   - Choosing anchors (Anchor-X and Anchor-Y) introduces variability that might affect system scores. The study could benefit from exploring alternative anchor design strategies or justifying their current selection in more detail.

5. **Potential Overhead in Combined Variant:**
   - The **MUSHRA-DG-NMR** variant combines the strengths of both approaches but might also inherit their weaknesses, such as increased time complexity and potential subjectivity.

6. **Bias Considerations:**
   - While the work addresses reference-matching bias and judgment ambiguity, other potential biases (e.g., cultural or gender-related influences on rating) remain unexplored.

### Suggestions for Improvement

1. Extend the study to include a broader range of languages, particularly high-resource languages like English, to validate findings across diverse linguistic contexts.
2. Provide a deeper discussion on the scalability of **MUSHRA-DG**, particularly in large-scale evaluation setups.
3. To enhance flexibility, Consider offering adaptive or task-specific scoring formulas in **MUSHRA-DG**.
4. Explore and evaluate additional anchor design methodologies to ensure evaluation robustness and consistency.
5. Investigate other potential biases in human evaluations, such as demographic or cultural factors, to further enhance fairness and reliability.

---

> ### Author Response · Authors · 2025-01-30
> **Response to Reviewer 1BMu**
>
> We sincerely thank Reviewer 1BMu for their thoughtful and constructive feedback. We are delighted to learn that the reviewer found our analysis to be comprehensive, the proposed variants to be innovative, the validation methodology thorough, and MANGO to be a valuable dataset underscoring the practical relevance of our work. We also appreciate the clear and actionable suggestions, which we address in detail below.
>
> ## Critical Changes
>
> ## **1. Generalizability Across Languages**
>
> The primary focus of our work is on low-resource languages, and our experiments span two diverse languages from distinct language families (viz., Indo-Aryan and Dravidian) with a combined speaker base of 562 million speakers. This diversity reflects our intent to ensure generalizability across language families.  We note that extending the analysis required additional resources, as human evaluations are both time and cost intensive. However, we acknowledge the reviewer's concern and agree that expanding the scope to include a high-resource language like English would strengthen our conclusions.
>
> To address this, we conducted additional experiments in English. We conduct the **MUSHRA** and **MUSHRA-DG-NMR** tests on the same set of systems trained on the LJSpeech dataset, with the help of 30 raters who rate 30 utterances each. All raters were native US English speakers balanced across age (18-60) and gender. We also conduct a CMOS test of all systems against the reference with 15 participants rating 30 utterances each.
> We present our findings and key insights from these new results, which strongly align with our earlier observations in Hindi and Tamil.
>
> ### **MUSHRA & MUSHRA-DG-NMR Scores (English)**
>
> | **System** | **MUSHRA** | **MUSHRA-DG-NMR** |
> |------------|:---------:|:----------------:|
> | ANC       |   37.23   |       50.16      |
> | FS2       |   47.96   |       59.23      |
> | VITS      |   69.55   |       72.56      |
> | ST2       |   72.63   |       74.32      |
> | REF       |   81.88   |       79.32      |
>
> ### **CMOS (English)**
>
> | **System** | **Mean (µ)** | **Std. Dev (σ)** | **95% CI** |
> |------------|:-----------:|:--------------:|:---------:|
> | REF       |      -      |       -        |     -     |
> | FS2       |   -0.78     |      2.39      |   0.22    |
> | VITS      |    0.02     |      1.25      |   0.12    |
> | ST2       |    0.21     |      1.25      |   0.12    |
>
> ### **Observations**
>
> **1. Is MUSHRA Reliable?**
>
> A key motivation for our work was to examine whether MUSHRA reliably reflects the actual perceptual differences between systems. Our new results in English reinforce our previous findings.  The reference (REF) is the only system that falls in the “Excellent” bin, while all other systems remain in the “Good” or “Fair” bin. However, the CMOS results tell a different story. CMOS indicates that ST2 and VITS are comparable to (if not better than) the reference, but MUSHRA fails to reflect this.
>
> This further confirms that reference-matching bias (RMB) is present in English as well. Raters subconsciously align their ratings to the reference, which distorts the perceived differences among systems. This supports our argument that MUSHRA scores can be misleading.
>
> **2. How Reliable is the Mean Statistic?**
>
> We further investigated the reliability of the mean statistic in MUSHRA using individual rater responses. Our analysis (see Figure -https://anonymous.4open.science/api/repo/ljspeech-evaluations-5086/file/figures/en_mushra_boxplot.png?v=03b9bbb7) reveals two major issues:
> * *Large variance in individual ratings:* A single rater does not always rate a system consistently across different utterances.
> * *High variance across raters:* Looking at the means of different raters, we see large differences in how they perceive and apply MUSHRA labels. This suggests judgment ambiguity—different raters interpret the MUSHRA scale differently, leading to inconsistent scoring across evaluators.
>
> These findings reinforce our claim that the mean statistic in MUSHRA alone is not a reliable indicator of perceived quality, as it is influenced by both rater inconsistency and individual interpretation of the scale.

---

> > ### Author Response · Authors · 2025-01-30
> > **Continuation of Response to Reviewer 1BMu [Critical Changes]**
> >
> > **3. Does Our Proposed Variant (MUSHRA-DG-NMR) Help?**
> >
> > Our proposed MUSHRA-DG-NMR test was designed to reduce reference-matching bias (RMB) and judgment ambiguity. The English results provide additional confirmation of its benefits:
> >
> > * MUSHRA-DG-NMR brings system scores closer to the reference, which indicates that RMB is mitigated. Raters are no longer as strongly influenced by the reference, leading to a more balanced evaluation of the systems.
> > * Additionally, MUSHRA-DG-NMR preserves the ranking of systems while shifting scores, just as we observed in Hindi and Tamil. This is an important feature, as it shows that MUSHRA-DG-NMR does not distort system rankings but makes scores more reliable.
> > * While we do not observe a significant reduction in score variance in English, one of the strongest advantages of our proposed evaluation framework is its ability to provide fine-grained insights beyond a single score. This is particularly useful when systems receive similar MUSHRA scores but have perceptible differences in quality. By analyzing both subjective (https://anonymous.4open.science/api/repo/ljspeech-evaluations-5086/file/figures/mushra-dg/subjective.png?v=af275cdc) and objective (https://anonymous.4open.science/api/repo/ljspeech-evaluations-5086/file/figures/mushra-dg/objective.png?v=2ee3d736) dimensions, we can pinpoint the specific factors contributing to listener preferences. For instance, VITS exhibits nearly twice as many sudden energy variations as ST2, while ST2 has nearly double the digital artifacts of VITS. Despite this, ST2 holds a perceptual advantage due to its superior rhythmic consistency, making it the preferred choice in terms of overall listening experience. These insights go beyond overall scores and help researchers identify and isolate faults in synthetic speech.
> >
> > **4. Final Thoughts**
> >
> > To further strengthen our study and its impact on the community, we will release all English ratings, expanding our annotation dataset to *255,000* subjective annotations. A dedicated section on our English findings has been included in the revised manuscript, along with supplementary figures in the Appendix. These additional English experiments have further reinforced our observations in Hindi and Tamil, strengthening our key conclusions:
> >
> > 1. MUSHRA scores do not always reflect true perceptual quality due to reference-matching bias
> > 2. Mean scores can be unreliable due to rating inconsistencies and judgment ambiguity
> > 3. Our MUSHRA-DG-NMR variant effectively mitigates bias while preserving system rankings
> > 4. Fine-grained analysis provides deeper insights beyond a single score.
> >
> > These findings further support our conclusions and confirm the robustness of our proposed methods across both high and low resource languages. We appreciate the reviewer’s valuable feedback and look forward to presenting these expanded results in our revised manuscript.
> >
> >
> > ## **2. Anchor Design Justification**
> >
> > Anchor-X was chosen as it adheres to the standard practice established in prior MUSHRA studies, ensuring consistency with existing benchmarks. Anchor-Y, on the other hand, was proposed as an ablation to analyze the impact of varying anchor design.
> >
> > * **Anchor-X:** This anchor was generated by downsampling high-fidelity audio to 3.5 kHz, effectively removing frequencies above this threshold, and then upsampling it back to 24 kHz, resulting in a signal with limited frequency content but standard sampling rate. This process serves as a baseline for evaluating systems, as it simulates the quality degradation typical in narrowband communication systems. It is widely accepted in the literature and aligns with prior work on MUSHRA evaluations. We re-emphasise that this was not newly proposed as a part of this work but is a standard anchor used in multiple prior works.
> >
> > * **Anchor-Y:** This was introduced solely for ablation purposes to assess the role of anchor design on evaluation outcomes.
> >
> > We emphasize that our main findings—(i) that MUSHRA without a human reference mitigates reference bias, and (ii) that MUSHRA with detailed guidelines improves reliability and reduces judgment ambiguity—remain valid regardless of the anchor used (Anchor-X in Hindi or Anchor-Y in Tamil). Hence, our use of Anchor-Y for Tamil serves as an exploratory addition as opposed to a mandatory recommendation, and does not affect the robustness of our conclusions.

---

> > > ### Author Response · Authors · 2025-01-30
> > > **Continuation of Response to Reviewer 1BMu [Critical Changes]**
> > >
> > > ## **3. Scalability Discussion for MUSHRA-DG**
> > >
> > > As highlighted in the paper, MUSHRA-DG and its variant MUSHRA-DG-NMR do require additional time for execution compared to standard MUSHRA tests. However, this added effort yields significant benefits by providing granular and actionable insights that are highly valuable in TTS system development.
> > >
> > > For instance:
> > > * If the test shows that prosody errors are more prevalent then it allows researchers to prioritize improvements in aligners or collect more expressive, conversational data rather than read speech.
> > > * Similarly, if the test shows that pronunciation errors are more prevalent then it emphasizes the need for better codec architectures or enhanced vocabulary coverage through targeted data collection.
> > >
> > > These insights guide development in a focused manner, potentially reducing the overall time required for system iteration and improvement. Thus, the additional time investment for MUSHRA-DG evaluations is justified by the downstream acceleration in development cycles and potentially reduced compute costs.
> > >
> > > We also acknowledge the reviewer’s suggestion to parallelize by exploring platforms like **Prolific** for parallelizing listener participation, thereby reducing the evaluation time. We have now integrated such platforms into our methodology for future experiments to enhance scalability.
> > >
> > > ## **4. Subjectivity in Scoring Formula**
> > >
> > > We understand the reviewer's concern regarding the subjective nature of the formula used to compute final scores in MUSHRA tests with detailed scoresheets. To address this, we wish to clarify that we developed the scoring formula through multiple discussions with raters and researchers. These conversations helped us design a scoring method that aligns with how raters typically evaluate TTS systems, making it closer to real-world rating practices. The formula combines perceptual measures, such as liveliness, voice quality, and rhythm, with penalties for observable errors like pronunciation mistakes, unnatural speed changes, and digital artifacts.
> > >
> > > The axes and weights in the formula were chosen based on feedback from these discussions. We observed that raters often focus on penalizing mistakes rather than explicitly rewarding correct features, which influenced how we designed the scoring process. We also considered the severity and impact of different types of errors during these conversations to create a formula that reflects what most raters prioritize, while allowing for some variation in individual preferences.
> > >
> > > While we acknowledge that this methodology may benefit from further refinement in future iterations, it represents a systematic, empirically grounded, and practically validated framework. By balancing perceptual qualities and penalties for errors, we believe this formulation reflects the rigor of our approach and provides a reasonable and useful scoring mechanism. We hope this explanation addresses the reviewer's concerns.

---

> > > > ### Author Response · Authors · 2025-01-30
> > > > **Continuation of Response to Reviewer 1BMu [Suggested Enhancements]**
> > > >
> > > > ## Suggested Enhancements
> > > >
> > > > **1. Bias Analysis:**
> > > >
> > > > We humbly acknowledge that investigating additional bias evaluations, such as cultural and demographic influences, is beyond the scope of the current work. While we recognize the significance of these factors, our focus is specifically on addressing biases intrinsic to the MUSHRA implementation in TTS evaluations. These broader biases are applicable across all existing TTS evaluation methods and merit a dedicated study to ensure thorough and meaningful analysis.
> > > >
> > > > **2. Visualization Improvements:**
> > > >
> > > > We have significantly improved the clarity and interpretability of all visualizations by updating captions and legends. The revised legends are now more descriptive, and the captions provide detailed explanations of the key data aspects and insights these visualizations convey. These enhancements are designed to ensure that even readers unfamiliar with the dataset can easily understand the results presented.
> > > >
> > > > **3. Dataset Accessibility:**
> > > >
> > > > To improve the accessibility of the MANGO dataset, we plan to release it on HuggingFace. This release will feature a dataset viewer, enabling researchers to explore specific entries, example ratings, and summaries. These additions aim to showcase the dataset's value and provide practical insights into its structure and utility for the research community.
> > > >
> > > > **4. Practical Applicability:**
> > > >
> > > > We sincerely appreciate the reviewer’s acknowledgment of our work as comprehensive, innovative, and robust. Achieving this has required significant human and monetary effort. We are eager to see our work applied to real-world use cases and have structured our study to closely mimic real-world settings. By publishing these findings, we aim to provide a strong foundation for practical adoption. We encourage the community to experiment with this framework and share their insights from real-world applications.
> > > >
> > > > **5. Comparative Analysis with Other Tests:**
> > > >
> > > > We reiterate the limitations of MOS tests, as discussed in the Introduction. MOS has been widely criticized for its lack of robustness and scalability, making it less meaningful as a gold standard for evaluation. Instead, we rely on CMOS tests as a reliable reference. Throughout our study (see Sections 4.1, 4.4, 6.1, 6.2, and 6.3), we compare MUSHRA and its proposed variants with CMOS in terms of rank consistency and score similarities, recognizing the different scales on which these tests operate. While CMOS is reliable, it becomes cumbersome to scale with many systems, which motivates our exploration of MUSHRA-based evaluations.
> > > >
> > > > **6. Future Research Directions:**
> > > >
> > > > Thank you for this valuable suggestion. We have added a dedicated section on future research directions. Potential avenues for exploration include:
> > > > * Adapting MUSHRA-DG-NMR for tournament-style evaluations, where new systems are incrementally added, necessitating a dynamic leaderboard.
> > > > * Incorporating automatic metrics alongside human evaluations to provide complementary perspectives.
> > > > * Streamlining MUSHRA for greater accessibility in large-scale studies.
> > > > * Exploring the integration of cultural and demographic bias analyses within the MUSHRA framework.
> > > >
> > > > These directions aim to further enhance the utility and applicability of MUSHRA-based evaluations in TTS research.
> > > >
> > > > We sincerely thank the reviewer once again for their invaluable feedback, which has significantly contributed to improving the quality of our work. We hope these revisions adequately address the concerns. We will upload the revised manuscript once all the reviews are in. Please let us know if any further clarifications are required.
> > > >
> > > > Sincerely,
> > > >
> > > > Authors

---

### Review · Reviewer_e65S · 2025-02-21

**Summary Of Contributions:**

This article conducts a large-scale MUSHRA listening test to gain an understanding of its features and limitations and examines two variants that better match authors expectations of an ideal listening test.

**Audience:**

Yes

**Broader Impact Concerns:**

I can think of none

**Claims And Evidence:**

Yes

**Requested Changes:**

Please address the following critical required changes (copied from above with small changes):

- lack of a dedicated section providing a good overview of MUSHRA, work done on improving this listening protocol and comparison to other kinds of listening test; some of this information is present in the paper but it is scattered making for a very confusing presentation
- I struggle to understand why reference matching is presented as a shortcoming of MUSHRA. This is exactly what MUSHRA appears to be measuring. Changing that aspect leads to a different test that combines features of multiple listening tests
- lack of clarity on the number of participants and ratings provided by each participant; one part of the paper says that 226,000 rating were collected; another part says that each participant provided 100 ratings; this suggests that there were 2260 participants; however, table 1 (left) says that the total number of ratings was 246,000 and the total number of participants is 471 and table 1 (right) suggests yet another number of participants.
- subjective statements regarding participant on-boarding: "were clearly explained", "any doubts were addressed"
- clarify why no exit interview with participants were performed to get insights into their decision making, why instead such insights were speculated
- naturalness, quality and other ambiguous concepts not clearly defined
- imprecise statements about Table 1: only some boxes reflect high variance, a substantial number of boxes has low variance
- Anchor-Y TTS model was never introduced
- address speculative statements regarding true opinions based on 1.3% ratings revised; the lack of incentive to revise puts that assertion to question; only exit interview could have corroborated that to some extent
- informal language like prowess
- error rates higher than 100% in Figure 5 need to be explained

**Strengths And Weaknesses:**

Key strengths:
- this is a large scale study conducted across 2 languages from 2 different language families and a number of modern TTS architectures
- it explores several variants of the standard MUSHRA test
- it measure several aspects of MUSHRA (reliability, sensitivity, anchor)

Key weaknesses:
- lack of a dedicated section providing a good overview of MUSHRA, work done on improving this listening protocol and comparison to other kinds of listening test; some of this information is present in the paper but it is scattered making for a very confusing presentation
- I struggle to understand why reference matching is presented as a shortcoming of MUSHRA. This is exactly what MUSHRA appears to be measuring. Changing that aspect leads to a different test that combines features of multiple listening tests
- lack of clarity on the number of participants and ratings provided by each participant; one part of the paper says that 226,000 rating were collected; another part says that each participant provided 100 ratings; this suggests that there were 2260 participants; however, table 1 (left) says that the total number of ratings was 246,000 and the total number of participants is 471 and table 1 (right) suggests yet another number of participants.
- subjective statements regarding participant on-boarding: "were clearly explained", "any doubts were addressed"
- no exit interview with participants were performed to get insights into their decision making, instead such insights were speculated
- naturalness, quality and other ambiguous concepts not clearly defined
- imprecise statements about Table 1: only some boxes reflect high variance, a substantial number of boxes has low variance
- Anchor-Y TTS model was never introduced
- speculative statements regarding true opinions based on 1.3% ratings revised; the lack of incentive to revise puts that assertion to question; only exit interview could have corroborated that to some extent
- informal language like prowess
- error rates higher than 100% in Figure 5 need to be explained

---

> ### Author Response · Authors · 2025-02-27
> **Response to Reviewer e65S**
>
> We sincerely thank Reviewer e65S for their thoughtful and constructive feedback.We appreciate the reviewer’s thoughtful feedback and recognition of our study’s strengths, including its large scale across two diverse languages, exploration of MUSHRA test variants, and analysis of reliability, sensitivity, and anchoring. We also appreciate the constructive suggestions, which we carefully address to strengthen key aspects of the study.
>
> # Critical Changes
>
> ## 1. Background on MUSHRA:
> * In Section 3, we introduce MUSHRA as a key evaluation method, describe its implementation with explicit references and anchors, and explain the rating scale. For detailed instructions provided to participants, we refer readers to Appendix A.1.
> * However, we recognize that the background, comparisons to prior tests, and work on improving the protocol may not have been as immediately clear as intended, and we are happy to refine this section for better readability.
> * We will revise Section 3 with the following details -
>
>    * **Background & Implementation:** To evaluate TTS systems, we adopt the standard MUSHRA test (ITU-R, 2015), a methodology originally developed for codec evaluation and widely used in speech and audio quality assessment. MUSHRA is particularly suited for listeners with normal hearing and experience in critical listening, as it requires evaluating multiple stimuli side by side while referencing a high-quality benchmark. Raters assess multiple stimuli per page, including an explicit reference (a high-quality speech benchmark), an anchor (a lower-quality version for calibration), and an implicit (hidden) reference to normalize judgments. Each stimulus is rated on a continuous 0–100 scale, discretized into five categories: 100-80 (Excellent), 80-60 (Good), 60-40 (Fair), 40-20 (Poor), and 20-0 (Bad). To ensure reliable and consistent evaluations, MUSHRA follows established guidelines according to the standard. Listeners must complete pre-test training to familiarize themselves with impairments and test signals. Stimuli should be around 10 seconds long, with a maximum of 12 seconds, to minimize listener fatigue and enhance response stability. An assessor’s responses are excluded if they rate the hidden reference below 90 in more than 15% of test items. Tests must use either headphones or loudspeakers, but not both within a session, ensuring consistency across participants. When listening conditions are technically and behaviorally controlled, data from as few as 20 subjects can yield statistically reliable conclusions. These measures help maintain the rigor and reproducibility of MUSHRA-based speech evaluations, making it a robust choice for large-scale TTS assessment.
>    * **Comparison:** MUSHRA offers several advantages over Mean Opinion Score (MOS) and Comparative MOS (CMOS) for speech quality assessment. MOS, despite its widespread use, has been criticized for its poor reliability and inability to distinguish between similar-sounding systems due to its single-stimulus nature and the absence of an explicit reference. While CMOS provides pairwise comparisons, it is inherently limited to evaluating only two systems at a time, making large-scale assessments both expensive and difficult to scale. In contrast, MUSHRA enables simultaneous comparisons across multiple systems, allowing listeners to switch between samples and directly compare quality, which enhances sensitivity to subtle differences that MOS often fails to capture. The continuous 0–100 scale provides finer granularity than MOS’s coarse 5-point rating, allowing for more nuanced judgments. Additionally, MUSHRA is more efficient than exhaustive pairwise CMOS tests, as it allows multiple systems to be evaluated in a single trial rather than requiring numerous pairwise comparisons. Another key strength is reference-based calibration, where listeners rate systems relative to a known high-quality reference and an anchor, with the aim of reduce scoring variability. Ribeiro et al. (2015) further confirm MUSHRA’s advantages, demonstrating that it more effectively distinguishes between models compared to MOS, reinforcing its suitability for evaluating modern TTS systems.

---

> > ### Author Response · Authors · 2025-02-27
> > **Continuation of Response to Reviewer e65S**
> >
> > * **Evolution:** Over time, researchers have adapted and refined the MUSHRA framework to better align with the evolving demands of speech synthesis evaluation. While the standard MUSHRA test remains widely used, several variations have emerged, modifying key aspects such as reference anchoring and test structure to address specific challenges in assessing TTS quality. For instance, BaseTTS [2] follows the MUSHRA framework but omits anchors, suggesting that explicit anchoring may not always be necessary for reliable judgments. NaturalSpeech2 [3] further questions the necessity of strict reference matching by employing a MUSHRA-like CMOS setup without an explicit reference. Additionally, Expressive-MUSHRA has been introduced to evaluate expressiveness in speech synthesis [4, 5], highlighting MUSHRA’s flexibility in capturing prosodic variation and expressive nuances. Beyond these structural modifications, studies have also examined MUSHRA’s effectiveness and limitations. For example, Merritt et al.[6] critique the traditional 3.5 kHz anchor, arguing that it may not be ideal for modern speech synthesis, as distortions in neural TTS systems differ from those originally considered in codec evaluations. These adaptations reflect a growing recognition that MUSHRA, while valuable, may require refinements to remain optimally effective for contemporary TTS evaluation. Recognizing this need, we systematically explore alternative test designs and their impact on evaluation reliability.
> >
> >
> > [1] Ribeiro, M. S., Yamagishi, J., & Clark, R. A. (2015). A perceptual investigation of wavelet-based decomposition of f0 for text-to-speech synthesis. In INTERSPEECH 2015 16th Annual Conference of the International Speech Communication Association (pp. 1586-1590). International Speech Communication Association.
> >
> > [2] Łajszczak, M., Cámbara, G., Li, Y., Beyhan, F., Van Korlaar, A., Yang, F., ... & Drugman, T. (2024). Base tts: Lessons from building a billion-parameter text-to-speech model on 100k hours of data. arXiv preprint arXiv:2402.08093
> >
> > [3] Shen, K., Ju, Z., Tan, X., Liu, Y., Leng, Y., He, L., ... & Bian, J. (2023). Naturalspeech 2: Latent diffusion models are natural and zero-shot speech and singing synthesizers. arXiv preprint arXiv:2304.09116.
> >
> > [4] Huybrechts, G., Merritt, T., Comini, G., Perz, B., Shah, R., & Lorenzo-Trueba, J. (2021, June). Low-resource expressive text-to-speech using data augmentation. In ICASSP 2021-2021 IEEE International Conference on Acoustics, Speech and Signal Processing (ICASSP) (pp. 6593-6597). IEEE.
> >
> > [5] Varadhan, P. S., Sankar, A., Raju, G., & Khapra, M. M. (2024). Rasa: Building Expressive Speech Synthesis Systems for Indian Languages in Low-resource Settings. In INTERSPEECH 2024, Kos Island, Greece, International Speech Communication Association.
> >
> > [6] Merritt, T., Putrycz, B., Nadolski, A., Ye, T., Korzekwa, D., Dolecki, W., ... & Barra-Chicote, R. (2018, December). Comprehensive evaluation of statistical speech waveform synthesis. In 2018 IEEE Spoken Language Technology Workshop (SLT) (pp. 325-331). IEEE.

---

> > > ### Author Response · Authors · 2025-02-27
> > > **Continuation of Response to Reviewer e65S**
> > >
> > > ## 2. Why Reference-Matching is a Shortcoming? Why do we propose a new variant?
> > >
> > >
> > > We appreciate the reviewer’s perspective but our concern lies in the assumption that the reference always represents the highest quality standard, which may not hold in practice. While MUSHRA relies on reference matching, we argue that this assumption may not always be valid, particularly in cases where TTS systems are perceived as superior to the reference itself. If a system frequently outperforms the human reference in preference ratings, the MUSHRA framework may unintentionally penalize the system by anchoring scores to a suboptimal reference.
> > >
> > >
> > > To better understand this effect, we convert CMOS scores to preference ratings, where CMOS > 0 indicates that a system is preferred over the reference, while CMOS < 0 means the reference is preferred. Our analysis reveals that StyleTTS2 is preferred over the reference in 54% of cases in Tamil and 43.8% in English, clearly demonstrating that model outputs can surpass human recordings. Even VITS achieves a preference rate of 37.2% in English, further supporting the argument that using the human reference as a fixed standard may not always be appropriate. Similar findings have been reported in other papers. For example, authors of StyleTTS2 show that their model surpasses human recordings in a CMOS test. The preference scores from our study are summarized in the table below. This raises concerns about whether the reference should always be the fixed standard in MUSHRA-based evaluations.
> > >
> > > | Language | System | Reference (%) | Equal (%) | System (%) |
> > > |----------|--------|--------------|-----------|------------|
> > > | English  | ST2    | 39.1         | 17.1      | **43.8**       |
> > > |          | VITS   | 38.7         | 24.1      | 37.2       |
> > > |          | FS2    | 68.3         | 13.5      | 18.2       |
> > > | Hindi    | ST2    | 42.8         | 18.0      | 39.2       |
> > > |          | VITS   | 43.9         | 18.2      | 37.8       |
> > > |          | FS2    | 59.9         | 11.8      | 28.3       |
> > > | Tamil    | ST2    | 41.0         | 5.0       | **54.0**       |
> > > |          | VITS   | 63.6         | 4.8       | 31.5       |
> > > |          | FS2    | 62.9         | 3.5       | 33.5       |
> > >
> > >
> > > These findings suggest that strict reference matching in MUSHRA may not always be the most reliable approach for evaluating modern TTS systems. To address this, we systematically explore test designs without an explicit reference (MUSHRA-NMR), assessing their impact on evaluation reliability. While our approach draws inspiration from multiple listening tests, our objective remains to refine MUSHRA to better accommodate evolving speech synthesis models while preserving its core strengths.
> > >
> > >
> > > ## 3. Clarity on Number of Participants and Ratings
> > >
> > > We appreciate the reviewer’s attention to detail regarding the participant and rating statistics. We could not spot the numbers 471 and 226,000 mentioned explicitly in the paper. However, we did see that in Table 1, while the total number of participants is correctly reported as 492, the sum of participants across age and gender categories amounts to 471. This discrepancy arises because demographic details (age and gender) for 21 participants were unavailable, as they opted not to provide this information. We would like to clarify that this does not affect the total number of ratings or the participant distribution across test conditions. Each of the 492 participants evaluated 100 utterances per system across five TTS systems, contributing 500 ratings each. This results in a total of 246,000 human ratings, as correctly reported in the manuscript. All other figures in Table 1, including the distribution of participants across different test variants and languages, remain accurate. We appreciate the reviewer’s keen observation and will ensure that this discrepancy is addressed in the final version.
> > >
> > >
> > > ## 4. Subjective Statements on Participant Onboarding
> > > We acknowledge the reviewer’s concern regarding the phrasing of our participant onboarding process. To ensure assessors were familiar with the subjective test setup, as recommended by the standard, we conducted demo calls where participants were introduced to the rating interface and evaluation process. During these sessions, we walked them through a sample rating task, addressing any potential doubts in real time. Additionally, all instructions provided to participants are included in Appendix A1, ensuring full transparency. To improve clarity, we have refined our description in the manuscript to focus on the specific onboarding methods used rather than subjective phrasing.

---

> > > > ### Author Response · Authors · 2025-02-27
> > > > **Continuation of Response to Reviewer e65S**
> > > >
> > > > ## 5. Clarification on Exit Interviews
> > > >
> > > > We would like to clarify that our insights into participant decision-making are not speculative. We conducted a structured post-evaluation participant survey from 89 participants after the original MUSHRA test to systematically collect feedback on how raters made their judgments.
> > > >
> > > >
> > > > The survey results, [summarized in this Table](https://anonymous.4open.science/api/repo/ljspeech-evaluations-5086/file/figures/post_evaluation_survey.pdf ), provide direct evidence supporting our claims. First, reference-matching plays a strong role in participant decisions (55% preferred the system closer to the reference), but it is not absolute, as 18% favored the non-reference-matched system and 11% rated it based on general quality. This reinforces that strict reference dependence may not always be ideal. Second, we explicitly assessed judgment ambiguity, where 56% prioritized voice quality over pronunciation errors, while 33% preferred better pronunciation over voice quality, demonstrating that raters face trade-offs that are not always resolved by reference comparison alone. Additionally, 56% of participants reported encountering rating confusion at least a few times, confirming scoring ambiguity exists despite clear guidelines. Notably, a significant portion (58%) did not rate the reference as “Excellent”, with some even rating it as Fair (19%) or Poor (1%), further questioning the assumption that the reference is always the highest-quality standard. Lastly, 22% of participants found the MUSHRA test difficult or very difficult, suggesting that cognitive load may influence rating consistency. These results collectively support our argument that while MUSHRA is valuable, strict reference-matching and scoring assumptions may not always be ideal, reinforcing the need for alternative evaluation setups. We have further clarified these findings in the manuscript to ensure transparency.
> > > >
> > > > The findings from this survey directly informed our analysis, ensuring that our interpretations were grounded in participant responses rather than speculation. To further improve clarity, we will explicitly reference this survey in the appendix in the revised manuscript.
> > > >
> > > >
> > > > ## 6. Clarification on Ambiguous Concepts
> > > >
> > > > We appreciate the reviewer’s concern regarding the definitions of terms such as naturalness and quality. In the speech synthesis community, these terms are commonly used to describe human-like speech characteristics, though precise, universally agreed-upon definitions are often lacking. As seen in prior work (e.g., NaturalSpeech, StyleTTS2), these aspects are typically evaluated through subjective human perception, given their inherently listener-dependent nature. In line with this convention, our study assesses naturalness and quality based on perceptual ratings. To improve clarity, we will explicitly mention in the manuscript that these terms reflect standard TTS evaluation practices and are interpreted through listener judgments of speech realism and overall fidelity.
> > > >
> > > >
> > > > ## 7. Variance in Figure 1.
> > > >
> > > > In Figure 1, the data clearly demonstrates substantial variance across ratings. While some reference conditions may exhibit lower variance, the variance in system ratings remains high, reflecting the diverse perceptual judgments across different models. This suggests that even when references are stable, participants exhibit a wide range of preferences for TTS systems, reinforcing the need for robust evaluation methods. To make this clearer, we have ensured that the description in the manuscript explicitly highlights this distinction between reference and system variance. We appreciate the opportunity to clarify this and hope this explanation resolves the concern.
> > > >
> > > >
> > > > ## 8. Anchor-Y TTS model was never introduced
> > > >
> > > > We would like to clarify that Anchor-Y and its construction process are explicitly described in Section 4.5 of our manuscript. In this section, we detail the motivation, degradation methods applied to ST2 outputs, and expected rating behavior. Specifically, we outline the pitch averaging, diffusion step reduction, playback speed modification, and text-induced errors used to create an anchor expected to fall in the “Poor” or “Fair” category. Additionally, we analyze its impact in Table 2, showing that despite its low quality the relative rankings of other systems remain stable, reinforcing the argument that anchors may not always be necessary in MUSHRA evaluations. We kindly refer the reviewer to Section 4.5 for the full discussion.

---

> > > > > ### Author Response · Authors · 2025-02-27
> > > > > **Continuation of Response to Reviewer e65S**
> > > > >
> > > > > ## 9. Regarding Statement on Revision of Ratings
> > > > >
> > > > > We would like to respond to the reviewer's comments about speculation in our conclusions. We would like to clarify that the design of our test was not based on intuition but emerged from an iterative process informed by extensive discussions and prior evaluations conducted over the past six months across multiple research efforts in our group. These evaluations involved continuous engagement with raters, whose feedback helped refine various aspects of the test design. In particular, the formulation of the rating adjustment mechanism was guided by insights from a select group of experienced raters with demonstrated expertise in annotating TTS system outputs. While the evolution of these choices was shaped by ongoing discussions over several months, capturing this organic process in a structured manner within the paper is inherently challenging. However, we will clarify in the manuscript that our design decisions were grounded in empirical experience and sustained interactions rather than intuition alone.
> > > > >
> > > > > ## 10. Regarding Informal Language like Prowess
> > > > >
> > > > > We appreciate the reviewer’s careful attention to our writing. Our intent was to highlight the effectiveness and capability of the MUSHRA-DG test in fault isolation, and we have updated the phrasing accordingly to maintain clarity and a formal tone. The updated version reads, “Keeping this in mind, we subsequently discuss the effectiveness of the MUSHRA-DG test in fault isolation."
> > > > >
> > > > > ## 11. Error Rates in Figure 5
> > > > >
> > > > > We recognize the need to clarify that the error rates appear higher than 100% in Figure 5. This is due to the stacked bar chart representation, where multiple error categories are visualized cumulatively rather than as independent percentages. To avoid confusion, we have updated the figure caption to explicitly clarify this.
> > > > >
> > > > > ---
> > > > > We sincerely thank the reviewer once again for their invaluable feedback, which has significantly contributed to improving the quality of our work. We hope these revisions adequately address the concerns. Please let us know if further clarifications are required.

---

### Author Response · Authors · 2025-03-06
**Revised Manuscript Submission: A Summary of Response to Reviewer Comments**

**Dear Reviewers,**

We sincerely appreciate your thoughtful feedback and the time you have dedicated to reviewing our manuscript. Your insights have been instrumental in refining our work, and we have carefully addressed all critical revisions to improve clarity, completeness, and presentation. Below is a summary of the key revisions:

### **Reviewer e65S**
- Expanded the discussion on MUSHRA, including improvements and comparisons with other listening test protocols (**Section 3.1**).
- Clarified the limitations of reference-matching with supporting evidence from CMOS-derived preference scores (**Appendix, Section 4.1**).
- Resolved the apparent discrepancy in participant numbers in **Table 1** (**Section 3.4**).
- Revised subjective wording related to participant onboarding (**Section 3.4**).
- Documented insights from post-evaluation surveys (**Section 5, Appendix A.5**).
- Added discussion on the concept of "naturalness" (**Related Work**).
- Improved the description of variance in **Figure 1** for greater precision (**Section 4.2**).
- Clarified the statement on rating revisions (**Section 5**).
- Adjusted informal language for consistency with an academic tone (**Section 6.2**).
- Provided a clearer explanation of error rates exceeding 100% in **Figure 5** caption.

### **Reviewer 1BMu**
- Expanded discussion on the generalizability of findings beyond English (**Section 7**).
- Revised the discussion on subjectivity in the scoring formula (**Section 5**).
- Enhanced clarity in figures and captions (**Figures 1 & 5**).
- Added a comparative analysis of different listening tests (**Section 3.1**).
- Included a note on practical applications and future research directions (**Section 8**).

### **Reviewer y228**
- Revised the discussion on subjectivity in the scoring formula (**Section 5**).
- Included an analysis of objective evaluation metrics (**Section 4.7, Table 5**).


We truly appreciate the insightful comments from all reviewers, which have helped strengthen the manuscript.  We have carefully addressed all critical revisions. If there is anything that needs further clarification, please let us know and we would be happy to address it.

**Thank you,**

Authors

---

> ### Author Response · Authors · 2025-03-08
> **Official Comment by Authors**
>
> Dear Reviewers and Action Editor,
>
> We sincerely appreciate your time and valuable feedback on our submission. We have carefully addressed the concerns raised in the reviews and incorporated the necessary revisions. We greatly value the opportunity to engage in further discussion to ensure that our responses sufficiently address all concerns. Please let us know if there are any aspects that require further clarification or if there are any additional steps we should take from our side.
>
> Thank you for your time and consideration.
>
> Thank you,
>
> Authors

---

### Decision · Action_Editor_GKMc · 2025-04-15

**Recommendation:** Accept as is

**Comment:**

This work examines and enhances the MUSHRA test—a perceptual audio quality metric originally developed for evaluating audio compression. Through a series of experiments, the authors highlight two major limitations of the standard MUSHRA protocol: (1) evaluators often bias their ratings toward the explicitly provided reference, and (2) the use of a single rating score can lead to ambiguity in quality assessment. To mitigate these issues, the authors introduce two alternative MUSHRA configurations: one without an explicit reference and another with more detailed rating guidelines. Additionally, they present the MANGO dataset, designed for TTS evaluation.

All reviewers acknowledged the contribution and strengths of the paper. The authors have also thoroughly addressed the concerns raised during the reviewers during the authors-reviewers discussion period. Therefore, I recommend acceptance.

I also encourage the authors to incorporate all clarifications and additional results provided during the rebuttal into the final version, as these would significantly enhance the manuscript.

**Audience:**

Overall, since this paper focuses on an evaluation metric for speech generation, its primary audience is the speech and audio research community. One reviewer even noted that the work might be more appropriately published in a specialized speech/audio journal, such as IEEE/ACM Transactions on Audio, Speech, and Language Processing. Nevertheless, given that MUSHRA has become a widely adopted evaluation standard in generative speech, audio, and music models—which are also highly relevant to the broader AI community—I believe this work is suitable for publication in TMLR. To put it differently, had this metric been proposed for computer vision tasks, its relevance to TMLR would not have been questioned. Therefore, I recommend accepting the submission.

**Claims And Evidence:**

Claims and evidence are convincing and supported by experiments / references.